# Using Autodiff to Estimate Posterior Moments, Marginals and Samples

**Sam Bowyer**[1]  **Thomas Heap**[2]  **Laurence Aitchison**[2]

[1]School of Mathematics, University of Bristol
[2]Department of Computer Science, University of Bristol

## Abstract

Importance sampling is a popular technique in Bayesian inference: by reweighting samples drawn from a proposal distribution we are able to obtain samples and moment estimates from a Bayesian posterior over latent variables. Recent work, however, indicates that importance sampling scales poorly — in order to accurately approximate the true posterior, the required number of importance samples grows is exponential in the number of latent variables [Chatterjee and Diaconis, 2018]. Massively parallel importance sampling works around this issue by drawing $K$ samples for each of the $n$ latent variables and reasoning about all $K^n$ combinations of latent samples. In principle, we can reason efficiently over $K^n$ combinations of samples by exploiting conditional independencies in the generative model. Previous work only detailed how to compute an ELBO/marginal likelihood estimator by summing over all $K^n$ combinations. However, that work did not give an approach for computing other quantities of interest, namely posterior expectations, marginals and samples, as computing these quantities is far more complex. Specifically, these computations involve iterating forward (following the generative process), then iterating backward through the generative model. These backward traversals can be very complex, and require different backward traversals for each operation of interest. Our contribution is to exploit the source term trick from physics to entirely avoid the need to hand-write backward traversals. Instead, we demonstrate how to simply and easily compute all the required quantities — posterior expectations, marginals and samples — by differentiating through a slightly modified marginal likelihood estimator.

## 1 INTRODUCTION

Importance weighting allows us to reweight samples drawn from a proposal in order to compute expectations of a different distribution, such as a Bayesian posterior. However, importance weighting breaks down in larger models. To demonstrate this, Chatterjee and Diaconis [2018] considered a model with data, $x$, latent variables $z$ a true posterior, $P(z|x)$ and a proposal $Q(z)$. They showed that the number of samples required to accurately approximate the true posterior scales as $\exp(D_{\text{KL}}(P(z|x)||Q(z)))$. Problematically, we expect the KL divergence to scale with $n$, the number of latent variables. Indeed, if we have $n$ latent variables, and $P(z|x) = \prod_{i=1}^{n} P(z_i|x)$ and $Q(z) = \prod_{i=1}^{n} Q(z_i)$ are IID over those $n$ latent variables, then the KL-divergence is exactly proportional to $n$. Thus, we expect the required number of importance samples to be exponential in the number of latent variables, and hence we expect accurate importance sampling to be intractable in larger models.

To resolve this issue we use a massively parallel importance sampling scheme that in effect uses an exponential number of samples to compute posterior expectations, marginals and samples [Heap et al., 2023]. This involves drawing $K$ samples of each of the $n$ latent variables from the proposal, then individually reweighting all $K^n$ combinations of all samples of all latent variables. While reasoning about all $K^n$ combinations of samples might seem intractable, we should in principle be able to perform efficient computations by exploiting conditional independencies in the underlying probabilistic generative model. These conditional independencies can be depicted by drawing a graph with latent variables as the nodes, and dependencies as the edges; such models are often known as "probabilistic graphical models", or even just "graphical models" [Jordan, 2003, Koller and Friedman, 2009].

However, many computations that are possible in principle are extremely complex in practice, and that turns out to be the case here. While we should be able to perform this reasoning over $K^n$ latent variables using methods from the dis-

crete variable graphical model literature, this turned out to be less helpful than we had hoped because these algorithms involve highly complex backward traversals of the generative model. Worse, different traversals are needed for computing posterior expectations, marginals and samples, making a general implementation challenging. Our contribution is to develop a much simpler approach to computing posterior expectations, marginals and samples, which entirely avoids the need to explicitly write backwards computations. Specifically, we show that posterior expectations, marginals and samples can be obtained simply by differentiating through (a slightly modified) forward computation that produces an estimate of the marginal likelihood. The required gradients can be computed straightforwardly using modern autodiff, and the resulting implicit backward computations automatically inherit potentially complex optimizations from the forward pass.

## 2 BACKGROUND

**Bayesian inference.** In Bayesian inference, we have a prior, $P(z')$ over latent variables (sometimes in the statistical literature called parameters), $z' \in \mathcal{Z}$, and a likelihood, $P(x|z')$ connecting the latents to the data, $x$. Here, we use $z'$ rather than $z$ because we reserve $z$ for future use as a collection of $K$ samples (Eq. 3). Our goal is to compute the posterior distribution over latent variables conditioned on observed data,

$$P(z'|x) = \frac{P(x|z')P(z')}{\int dz'' \ P(x, z'')}, \tag{1}$$

We often seek to compute posterior expectations,

$$m_{\text{post}} = \int dz' \ P(z'|x) \, m(z') \tag{2}$$

but these are usually intractable, so instead we are forced to use an alternative method such as importance weighting.

**Importance weighting.** In importance weighting, we draw a collection of $K$ samples from the full joint latent space. A single sample is denoted $z^k \in \mathcal{Z}$, while the collection of $K$ samples is denoted,

$$z = (z^1, z^2, \ldots, z^K) \in \mathcal{Z}^K. \tag{3}$$

The collection of $K$ samples, $z$, is drawn by sampling $K$ times from the proposal,

$$Q(z) = \prod_{k \in \mathcal{K}} Q(z^k), \tag{4}$$

where $\mathcal{K}$ is the set of possible indices, $\mathcal{K} = \{1, \ldots, K\}$. As the true posterior moment is usually intractable, one approach is to use a self-normalized importance sampling estimate, $m_{\text{global}}(z)$. We call this a "global" importance weighted estimate following terminology in Geffner and

Domke [2022]. The global importance weighted moment estimate is,

$$m_{\text{global}}(z) = \frac{1}{K} \sum_{k \in \mathcal{K}} \frac{r_k(z)}{\mathcal{P}_{\text{global}}(z)} m(z^k) \tag{5}$$

where, samples, $z$, are drawn from the proposal (Eq. 4), and

$$r_k(z) = \frac{P(x, z^k)}{Q(z^k)} \tag{6}$$

$$\mathcal{P}_{\text{global}}(z) = \frac{1}{K} \sum_{k \in \mathcal{K}} r_k(z) \tag{7}$$

Here, $r_k(z)$ is the ratio of the generative and proposal probabilities, and $\mathcal{P}_{\text{global}}(z)$ is an unbiased estimator of the marginal likelihood,

$$\begin{aligned} E_{Q(z)}[\mathcal{P}_{\text{global}}(z)] &= E_{Q(z^k)}\left[\frac{P(x, z^k)}{Q(z^k)}\right] \\ &= \int dz^k \ P(x, z^k) = P(x) \quad (8) \end{aligned}$$

The first equality arises because $\mathcal{P}_{\text{global}}(z)$ is the average of $K$ IID terms, $P(x, z^k)/Q(z^k)$, so is equal to the expectation of a single term, and the second equality arises if we write the expectation as an integral.

**Source term trick.** Here, we outline a standard trick from physics that can be used to compute expectations of arbitrary probability distribution by differentiating a modified log-normalizing constant. This trick is used frequently in Quantum Field Theory, for instance [Weinberg, 1995] (Chapter 16), and also turns up in the theory of neural networks [Zavatone-Veth et al., 2021]. But the trick is simple enough that we can give a self-contained introduction here.

In our context, Bayes theorem (Eq. 1) defines an unnormalized density, $P(z|x) \propto P(x, z)$, with normalizing constant, $\int dz' \ P(x, z')$. Of course, the normalizing constant is usually intractable, but one of our contributions will be to show that the massively parallel estimate of the normalizing constant is sufficient to apply the source term trick. It turns out that we can compute posterior expectations using a slightly modified normalizing constant,

$$Z_m(J) = \int dz' \ P(x, z') e^{Jm(z')}. \tag{9}$$

where $e^{Jm(z')}$ is known as a source term, and $E_{P(z'|x)}[m(z')]$ is the moment we wish to compute. Note that setting $J$ to zero recovers the usual normalizing constant, $Z_m(J = 0) = \int dz' \ P(x, z')$. Now, we can extract the posterior moment by evaluating the gradient of

$\log Z_m(J)$ at $J = 0$,

$$\left.\frac{\partial}{\partial J}\right|_{J=0} \log Z_m(J)$$

$$= \left.\frac{\partial}{\partial J}\right|_{J=0} \log \int dz' \, \mathrm{P}(x, z') \, e^{Jm(z')}$$

$$= \frac{\int dz' \, \mathrm{P}(x, z') \left.\frac{\partial}{\partial J}\right|_{J=0} e^{Jm(z')}}{\int dz'' \, \mathrm{P}(x, z'')}.$$

Differentiating the exponential at $J = 0$ (first equality), and identifying the posterior using Bayes theorem (Eq. 1) (second equality),

$$= \int dz' \, \frac{\mathrm{P}(x, z')}{\int dz'' \, \mathrm{P}(x, z'')} m(z')$$

$$= \int dz' \, \mathrm{P}(z'|x) \, m(z) = m_{\mathrm{post}} \qquad (10)$$

We get back exactly the form for the posterior moment in Eq. (2).

**Massively parallel marginal likelihood estimators.** In the massively parallel setting, we assume a probabilistic graphical model with multiple latent variables $z_i' \in \mathcal{Z}_i$, indexed $i$. We can then form $z'$, used above, as a tuple containing all latent variables,

$$z' = (z_1', z_2', \ldots, z_n') \in \mathcal{Z}. \qquad (11)$$

We draw $K$ samples for each latent variable. We write the $k$th sample of the $i$th latent variable as $z_i^k$, so all $K$ samples of the $i$th latent variable can be written,

$$z_i = (z_i^1, z_i^2, \ldots, z_i^K) \in \mathcal{Z}_i^K. \qquad (12)$$

where $\mathcal{Z}_i$ is the space for the $i$th latent variable. And $z$ is the collection of all $K$ samples of all $n$ latent variables, (as in Eq. 3),

$$z = (z_1, z_2, \ldots, z_n) \in \mathcal{Z}^K. \qquad (13)$$

In the massively parallel setting, proposals have graphical model structure,

$$\mathrm{Q}_{\mathrm{MP}}(z) = \prod_{i=1}^{n} \mathrm{Q}_{\mathrm{MP}}(z_i | z_j \text{ for } j \in \mathrm{qa}(i)), \qquad (14)$$

where $\mathrm{qa}(i)$ is the set of indices of parents of $z_i$ under that graphical model. This massively parallel proposal over all copies of the $i$th latent variable, $z_i = (z_i^1, z_i^2, \ldots, z_i^K)$, arises from a user-specified, single-sample approximate posterior, $Q(z_i'|z_j' \text{ for } j \in \mathrm{qa}(i))$, where $z_i'$ and $z_j'$ are a single copy of the $i$th and $j$th latent variable. For instance, the massively parallel proposal might be IID over the K copies, $z_i^k$, and be based on a uniform mixture over all parent samples (other alternatives are available: see [Heap et al., 2023] for further details).

For the generative model, we need to explicitly consider all $K^n$ combinations of $K$ samples on $n$ latent variables. To help us write down these combinations, we define a vector of indices, $\mathbf{k}$, with one index, $k_i$ for each latent variable, $z_i$.

$$\mathbf{k} = (k_1, k_2, \ldots, k_n) \in \mathcal{K}^n, \qquad (15)$$

$$z^{\mathbf{k}} = \left(z_1^{k_1}, z_2^{k_2}, \ldots, z_n^{k_n}\right) \in \mathcal{Z}, \qquad (16)$$

That, allows us to write the "indexed" latent variables, $z^{\mathbf{k}}$, which represents a single sample from the full joint latent space. The generative model also has graphical model structure, with the set of indices of parents of the $i$th latent variable under the generative model begin denoted $\mathrm{pa}(i)$ (contrast this with $\mathrm{qa}(i)$ which is the parents of the $i$th latent variable under the proposal). The generative probability for a single combination of samples, denoted $z^{\mathbf{k}}$, can be written as,

$$\mathrm{P}(x, z^{\mathbf{k}}) = \mathrm{P}\left(x \middle| z_j^{k_j} \text{ for all } j \in \mathrm{pa}(x)\right)$$
$$\prod_{i=1}^{n} \mathrm{P}\left(z_i^{k_i} \middle| z_j^{k_j} \text{ for all } j \in \mathrm{pa}(i)\right). \qquad (17)$$

Thus, we can write a massively parallel marginal likelihood estimator as,

$$\mathcal{P}_{\mathrm{MP}}(z) = \frac{1}{K^n} \sum_{\mathbf{k} \in \mathcal{K}^n} r_{\mathbf{k}}(z) \qquad (18)$$

where

$$r_{\mathbf{k}}(z) = \frac{\mathrm{P}(x, z^{\mathbf{k}})}{\prod_i \mathrm{Q}_{\mathrm{MP}}\left(z_i^{k_i} \middle| z_j \text{ for } j \in \mathrm{qa}(i)\right)} \qquad (19)$$

The next challenge is to compute the sum in Eq. (18). The sum looks intractable as we have to sum over $K^n$ settings of $\mathbf{k}$. However, it turns out that these sums are usually tractable. The reason is that that if we fix the samples, $z$, then $r_{\mathbf{k}}(z)$ can be understood as a product of low-rank tensors,

$$r_{\mathbf{k}}(z) = f_{\mathbf{k}_{\mathrm{pa}(x)}}^x(z) \prod_i f_{k_i, \mathbf{k}_{\mathrm{pa}(i)}}^i(z) \qquad (20)$$

$$f_{\mathbf{k}_{\mathrm{pa}(x)}}^x(z) = \mathrm{P}\left(x \middle| z_j^{k_j} \text{ for all } j \in \mathrm{pa}(x)\right), \qquad (21)$$

$$f_{k_i, \mathbf{k}_{\mathrm{pa}(i)}}^i(z) = \frac{\mathrm{P}\left(z_i^{k_i} \middle| z_j^{k_j} \text{ for all } j \in \mathrm{pa}(i)\right)}{\mathrm{Q}_{\mathrm{MP}}\left(z_i^{k_i} \middle| z_j \text{ for all } j \in \mathrm{qa}(i)\right)}. \qquad (22)$$

Here, $f_{\mathbf{k}_{\mathrm{pa}(x)}}^x(z)$ is a tensor of rank $|\mathrm{pa}(x)|$, and $f_{k_i, \mathbf{k}_{\mathrm{pa}(i)}}^i(z)$ are tensors of rank $1 + |\mathrm{pa}(i)|$, where $|\mathrm{pa}(i)|$ is the number of parents of the $i$th latent variable. Thus, Eq. (18) is a large tensor product,

$$\mathcal{P}_{\mathrm{MP}}(z) = \frac{1}{K^n} \sum_{\mathbf{k} \in \mathcal{K}^n} f_{\mathbf{k}_{\mathrm{pa}(x)}}^x(z) \prod_i f_{k_i, \mathbf{k}_{\mathrm{pa}(i)}}^i(z) \qquad (23)$$

which can be efficiently computed using an opt-einsum implementation.

Now, we are in a position to define an importance sampling scheme that operates on all $K^n$ combinations of samples,

$$m_{\text{MP}}(z) = \frac{1}{K^n} \sum_{\mathbf{k} \in \mathcal{K}^n} \frac{r_{\mathbf{k}}(z)}{\mathcal{P}_{\text{MP}}(z)} m(z^{\mathbf{k}}). \qquad (24)$$

This looks very similar to the standard global importance sampling scheme in Eq. (5), except that Eq. (5) averages only over $K$ samples, whereas this massively parallel moment estimator averages over all $K^n$ combinations of samples. For a proof that this is a valid importance-sampled moment estimator, see Appendix section A.

# 3 METHODS

Of course, the contributions of this paper are not in computing the unbiased marginal likelihood estimator, which previously has been used in learning general probabilistic models, but instead our major contribution is a novel approach to computing key quantities of interest in Bayesian computation by applying the source term trick to the massively parallel marginal likelihood estimator. In particular, in the following sections, we outline in turn how to compute posterior expectations, marginals and samples.

**Interpreting massively parallel importance weighting as inference in a discrete variable graphical model.** Now, individual terms in (Eq. 24) can be understood as a normalized probability distribution over $\mathbf{k}$,

$$P_z(\mathbf{k}) = \frac{1}{K^n} \frac{r_{\mathbf{k}}(z)}{\mathcal{P}_{\text{MP}}(z)}. \qquad (25)$$

In particular, this quantity is always positive, and we can show that it normalizes to 1 by substituting the definition of $\mathcal{P}_{\text{MP}}(z)$ from Eq. (18),

$$\sum_{\mathbf{k}} P_z(\mathbf{k}) = \frac{1}{K^n} \sum_{\mathbf{k}} \frac{r_{\mathbf{k}}(z)}{\mathcal{P}_{\text{MP}}(z)} = \frac{\frac{1}{K^n} \sum_{\mathbf{k}} r_{\mathbf{k}}(z)}{\frac{1}{K^n} \sum_{\mathbf{k}'} r_{\mathbf{k}'}(z)} = 1 \qquad (26)$$

As such, we can in principle use methods for discrete variable graphical models, treating $\mathbf{k}$ as a random variable. However, as discussed in Related Work, computing posterior expectations, marginals and samples in discrete variable graphical models may still involves complex backward traversals, which are especially difficult if we want to exploit structure such as plates or timeseries to speed up the computations.

**Computing expectations by differentiating an estimate of the normalizing constant.** Instead, we modify our marginal likelihood estimator with a source term, $e^{Jm(z^{\mathbf{k}})}$,

$$\mathcal{P}_{\text{MP}}^{\text{exp}}(z, J) = \frac{1}{K^n} \sum_{\mathbf{k} \in \mathcal{K}^n} r_{\mathbf{k}}(z) e^{Jm(z^{\mathbf{k}})}. \qquad (27)$$

Remember that $r_{\mathbf{k}}(z)$ is a product of low-rank tensors, indexed by subsets of $\mathbf{k}$ (Eq. 20), so the sum can be computed efficiently using opt-einsum. Critically, the source term is just another factor with indices given by a subset of $\mathbf{k}$. For instance, most often $m$ (the function whose expectation we want to compute) will depend on only a single latent variable $m(z^{\mathbf{k}}) = m(z_i^{k_i})$, in which case the source term can be understood as just another tensor in the tensor product (Eq. 23), with one index, $k_i$. Now, we prove that differentiating the logarithm of this modified marginal likelihood estimator gives back a massively parallel moment estimator. In particular, we differentiate $\log \mathcal{P}_{\text{MP}}^{\text{exp}}(z, J)$ at $J = 0$ (first equality). Then in the numerator we substitute $\mathcal{P}_{\text{MP}}^{\text{exp}}(z, J)$ from Eq. (27), and in the denominator, we remember that $\mathcal{P}_{\text{MP}}^{\text{exp}}(z, J = 0) = \mathcal{P}_{\text{MP}}(z)$,

$$\frac{\partial}{\partial J}\bigg|_{J=0} \log \mathcal{P}_{\text{MP}}^{\text{exp}}(z, J)$$
$$= \frac{\frac{\partial}{\partial J}\big|_{J=0} \mathcal{P}_{\text{MP}}^{\text{exp}}(z, J)}{\mathcal{P}_{\text{MP}}^{\text{exp}}(z, 0)}$$
$$= \frac{\frac{1}{K^n} \sum_{\mathbf{k}} r_{\mathbf{k}}(z) \frac{\partial}{\partial J}\big|_{J=0} e^{Jm(z^{\mathbf{k}})}}{\mathcal{P}_{\text{MP}}(z)}$$

Computing the gradient of $e^{Jm(z^{\mathbf{k}})}$ at $J = 0$,

$$= \frac{\frac{1}{K^n} \sum_{\mathbf{k}} r_{\mathbf{k}}(z) m(z^{\mathbf{k}})}{\mathcal{P}_{\text{MP}}(z)} = m_{\text{MP}}(z) \qquad (28)$$

where the final equality comes from the definition of $m_{\text{MP}}(z)$ in Eq. (24). Note that this derivation is quite different from the standard "source-term trick" from physics described in Background, which works with either the true normalizing constant, or with a low-order perturbation to that normalizing constant. In contrast, here we use a very different massively parallel sample-based estimate of the marginal likelihood. Importantly, the subsequent two derivations are even more different from uses of the "source-term trick" in physics. In particular, the source-term trick is almost always used to compute moments/expectations in physics, whereas the subsequent two derivations use the same trick to compute quite different quantities (namely, probability distributions over samples).

Psuedocode for all procedures can be found in Appendix section B.

**Computing marginal importance weights.** Computing expectations directly is very powerful and almost certainly necessary for computing complex quantities that depend on multiple latent variables. However, if we are primarily interested in posterior expectations of individual variables, then it is considerably more flexible to compute "marginal" posterior importance weights. Once we have these marginal importance weights, we can easily compute arbitrary posterior expectations for individual variables, along with other quantities such as effective sample sizes. To define the marginal weights for the $i$th latent, note a moment for the

$i$th latent variable can be written as a sum over $k_i$,

$$m_{\text{MP}}(z) = \sum_{\mathbf{k} \in \mathcal{K}^n} \frac{r_{\mathbf{k}}(z)}{\mathcal{P}_{\text{MP}}(z)} m(z_i^{k_i}) = \sum_{k_i} w_{k_i}^i m(z_i^{k_i}), \quad (29)$$

where $w_{k_i}^i$ are the marginal importance weights for the $i$th latent variable, which are defined by,

$$w_{k_i}^i = \frac{\frac{1}{K^n} \sum_{\mathbf{k}/k_i \in \mathcal{K}^{n-1}} r_{\mathbf{k}}(z)}{\mathcal{P}_{\text{MP}}(z)}, \quad (30)$$

where the sum is over all $\mathbf{k}$ except $k_i$. Formally,

$$\mathbf{k}/k_i = (k_1, \ldots, k_{i-1}, k_{i+1}, \ldots, k_n) \in \mathcal{K}^{n-1}. \quad (31)$$

Again we can compute the marginal importance weights using gradients of a slightly different modified marginal likelihood estimator. Specifically, we now use a vector-valued $\mathbf{J} \in \mathbb{R}^K$ in a slightly different modified marginal likelihood estimator,

$$\mathcal{P}_{\text{MP}}^{\text{marg}}(z, \mathbf{J}) = \frac{1}{K^n} \sum_{\mathbf{k}} r_{\mathbf{k}}(z) e^{J_{k_i}}. \quad (32)$$

Again, $\mathcal{P}_{\text{MP}}^{\text{marg}}(z, \mathbf{0}) = \mathcal{P}_{\text{MP}}(z)$. As before, we differentiate $\log \mathcal{P}_{\text{MP}}^{\text{marg}}(z, \mathbf{J})$ at $\mathbf{J} = \mathbf{0}$,

$$\left. \frac{\partial}{\partial J_{k_i'}} \right|_{\mathbf{J}=\mathbf{0}} \log \mathcal{P}_{\text{MP}}^{\text{marg}}(z, \mathbf{J})$$

$$= \frac{\left. \frac{\partial}{\partial J_{k_i'}} \right|_{\mathbf{J}=\mathbf{0}} \mathcal{P}_{\text{MP}}^{\text{marg}}(z, \mathbf{J})}{\mathcal{P}_{\text{MP}}^{\text{marg}}(z, \mathbf{0})}. \quad (33)$$

Substituting for $\mathcal{P}_{\text{MP}}^{\text{marg}}(z, \mathbf{J})$ in the numerator,

$$= \frac{\frac{1}{K^n} \sum_{\mathbf{k}} r_{\mathbf{k}}(z) \left. \frac{\partial}{\partial J_{k_i'}} \right|_{\mathbf{J}=\mathbf{0}} e^{J_{k_i}}}{\mathcal{P}_{\text{MP}}(z)}. \quad (34)$$

The gradient is 1 when $k_i' = k_i$ and zero otherwise which can be represented using a Kronecker delta,

$$= \frac{\frac{1}{K^n} \sum_{\mathbf{k}} r_{\mathbf{k}}(z) \delta_{k_i', k_i}}{\mathcal{P}_{\text{MP}}(z)}. \quad (35)$$

We can rewrite this as a sum over all $\mathbf{k}$ except $k_i$,

$$= \frac{\frac{1}{K^n} \sum_{\mathbf{k}/k_i \in \mathcal{K}^{n-1}} r_{\mathbf{k}}(z)}{\mathcal{P}_{\text{MP}}(z)} = w_{k_i}^i,$$

which is exactly the definition of the marginal importance weights in Eq. (30).

**Computing conditional distributions for importance sampling.** A common alternative to importance weighting is importance sampling. In importance sampling, we

rewrite the usual estimates of the expectations in terms of a distribution over indices, $\mathrm{P}_z(\mathbf{k})$,

$$m_{\text{MP}}(z) = \sum_{\mathbf{k} \in \mathcal{K}^n} \mathrm{P}_z(\mathbf{k}) m(z^{\mathbf{k}}) \quad (36)$$

where

$$\mathrm{P}_z(\mathbf{k}) = \frac{1}{K^n} \frac{1}{\mathcal{P}_{\text{MP}}(z)} r_{\mathbf{k}}(z) \quad (37)$$

We can obtain (approximate) posterior samples, $z^{\mathbf{k}}$, by sampling $\mathbf{k}$ from $\mathrm{P}_z(\mathbf{k})$. However, sampling from $\mathrm{P}_z(\mathbf{k})$ is difficult in our context, as there are $K^n$ possible settings of $\mathbf{k}$, so we cannot explicitly compute the full distribution. Instead, we can factorise the distribution, and iteratively sample (e.g. we sample $k_1$ from $\mathrm{P}_z(k_1)$ then sample $k_2$ from $\mathrm{P}_z(k_2|k_1)$ etc). Formally, we use,

$$\mathrm{P}_z(\mathbf{k}) = \prod_i \mathrm{P}_z\left(k_i | \mathbf{k}_{\text{pa}(i)}\right) \quad (38)$$

where

$$\mathbf{k}_{\text{pa}(i)} = (k_j \text{ for all } j \in \text{pa}(i)) \quad (39)$$

where, remember pa$(i)$ is the set of indices of parents of the $i$th latent variable under the generative model, and $\mathbf{k}_{\text{pa}(i)}$ is the value of $k$ for each of those parents,

$$\mathbf{k}_{\text{pa}(i)} = (k_j \text{ for all } j \in \text{pa}(i)). \quad (40)$$

Note that this quantity is similar to the "backward kernels" in the SMC literature [Del Moral et al., 2006], what's different is our approach to computing the quantity, using the source-term trick to avoid the need for explicit backward traversals. Now, we have the problem of computing the conditionals, $\mathrm{P}_z\left(k_i | \mathbf{k}_{\text{pa}(i)}\right)$. We can compute the conditionals from the marginals using Bayes theorem,

$$\mathrm{P}\left(k_i | \mathbf{k}_{\text{pa}(i)}\right) = \frac{\mathrm{P}_z\left(k_i, \mathbf{k}_{\text{pa}(i)}\right)}{\sum_{k_i'} \mathrm{P}_z\left(k_i', \mathbf{k}_{\text{pa}(i)}\right)} \quad (41)$$

where

$$\mathrm{P}_z\left(k_i, \mathbf{k}_{\text{pa}(i)}\right) = \sum_{\mathbf{k}/(k_i, \mathbf{k}_{\text{pa}(i)})} \mathrm{P}_z(\mathbf{k}) \quad (42)$$

Again, we can compute these marginals efficiently by differentiating a modified estimate of the marginal likelihood. This time, we take a tensor-valued $\mathbf{J} \in \mathbb{R}^{K^{1+|\text{pa}(i)|}}$, where remember $|\text{pa}(i)|$ is the number of parents of the $i$th latent variable under the generative model.

$$\mathcal{P}_{\text{MP}}^{\text{samp}}(z, \mathbf{J}) = \frac{1}{K^n} \sum_{\mathbf{k}} \frac{\mathrm{P}(x, z^{\mathbf{k}})}{\mathrm{Q}(z^{\mathbf{k}})} e^{J_{k_i, \mathbf{k}_{\text{pa}(i)}}} \quad (43)$$

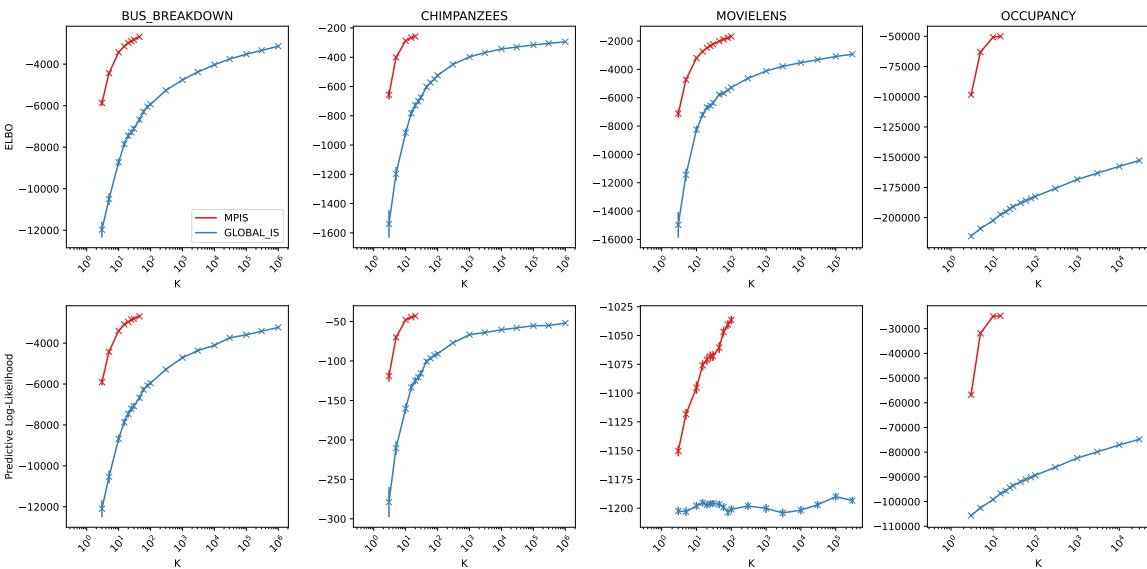

Figure 1: ELBO (top row) and predictive log-likelihood (bottom row) across the four datasets achieved via MP IS and global IS, for varying values of K. The error bars represent the standard-error across 100 repeated experiments on the same data but using different random seeds.

As usual, we differentiate with respect to $\mathbf{J}$ at $\mathbf{J} = \mathbf{0}$,

$$
\frac{\partial}{\partial J_{k_i', \mathbf{k}_{\mathrm{pa}(i)}'}}\bigg|_{\mathbf{J}=\mathbf{0}} \log \mathcal{P}_{\mathrm{MP}}^{\mathrm{samp}}(z, \mathbf{J})
$$

$$
= \frac{\frac{1}{K^n}\sum_{\mathbf{k}} r_{\mathbf{k}}(z)\delta_{(k_i, \mathbf{k}_{\mathrm{pa}(i)}),(k_i', \mathbf{k}_{\mathrm{pa}(i)}')}}{\mathcal{P}_{\mathrm{MP}}(z)}
$$

Here, $\delta_{(k_i, \mathbf{k}_{\mathrm{pa}(i)}),(k_i', \mathbf{k}_{\mathrm{pa}(i)}')}$ is a generalisation of the Kronecker delta. It is 1 when all the indices match (i.e. $k_i = k_i'$, and $\mathbf{k}_{\mathrm{pa}(i)} = \mathbf{k}_{\mathrm{pa}(i)}'$) and zero otherwise. These turn out to be precisely the marginals in Eq. (41),

$$
= \frac{\frac{1}{K^n}\sum_{\mathbf{k}/(k_i, \mathbf{k}_{\mathrm{pa}(i)})} r_{\mathbf{k}}(z)}{\mathcal{P}_{\mathrm{MP}}(z)}
$$

$$
= \sum_{\mathbf{k}/(k_i, \mathbf{k}_{\mathrm{pa}(i)})} \mathrm{P}_z(\mathbf{k}) = \mathrm{P}_z\left(k_i, \mathbf{k}_{\mathrm{pa}(i)}\right). \quad (44)
$$

## 4 EXPERIMENTS

We considered four datasets: NYC Bus Breakdown, Chimpanzee Prosociality, MovieLens100K and North American Breeding Bird Survey. NYC Bus Breakdown describes the length of around 150,000 delays to New York school bus journeys, segregated by year, borough, bus company and journey type. The Chimpanzee Prosociality dataset contains the actions taken in a controlled experiment by 7 chimpanzees given the repeated choice of whether to give food to another chimpanzee or not when receiving food themselves, repeated 12 times on 6 separate occasions per chimpanzee.

MovieLens100K [Harper and Konstan, 2015] contains 100k ratings of 1682 films from among 943 users. Finally, the North American Breeding Bird Survey records the number of sightings of over 700 bird species from 1966-2021 (excluding 2020), along thousands of different road-side routes in the United States and Canada. We use hierarchical probabilistic graphical models for these datasets, subsampling each dataset except for Chimpanzee Prosociality. For each dataset we define a generative model $\mathrm{P}\left(x, z^{\mathbf{k}}\right)$ and a factorised proposal $\mathrm{Q}\left(z^{\mathbf{k}}\right)$ which contains the same latents as the generative model, but each latent is independently parameterised by a simple (usually standard Normal) distribution. Note that we don't necessarily require our proposal $\mathrm{Q}\left(z^{\mathbf{k}}\right)$ to be factorised—this is done only for simplicity. The full details of these models are described in Appendix section C and a repository containing the code needed to reproduce these experiments can be found at `https://github.com/sambowyer/MPIS`.

We focus on two metrics to assess the quality of importance weighting and importance sampling: the importance weighted evidence lower bound (ELBO) and predictive log-likelihood. We use predictive log-likelihood to evaluate importance sample quality by drawing latent samples conditioned on observed 'training' data and use these to predict unobserved 'test' data. Therefore a higher predictive log-likelihood would correspond to higher-quality sampled latents (which are closer to the true posterior). The quality of importance weighting is measured by the tightness of the importance weighted ELBO, a bound on the model evidence. This is widely accepted as a good proxy for the quality of

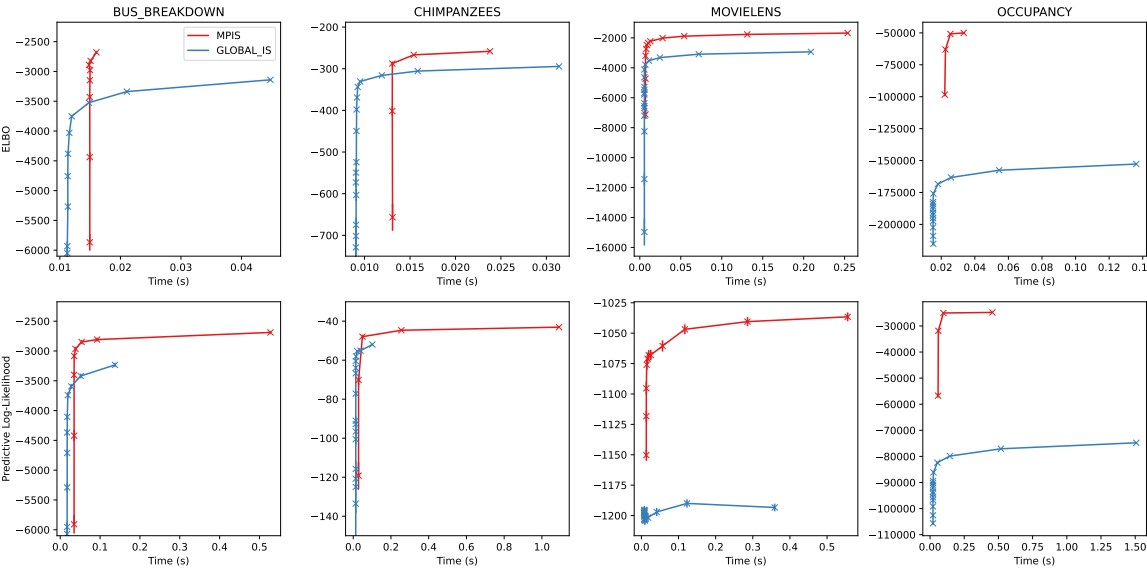

Figure 2: Results analogous to those in Fig. 1 but with time on the x-axis. Again, the error bars represent standard errors over 100 runs on the same data but using different random seeds.

the importance weighted posterior estimate [Geffner and Domke, 2022, Agrawal and Domke, 2021], as was confirmed empirically in Domke and Sheldon [2019]. This is because it can be interpreted as a single-sample ELBO with an improved approximate posterior [Cremer et al., 2017, Bachman and Precup, 2015], and the single-sample ELBO directly measures the discrepancy between the true and approximate posterior:

$$\text{ELBO} = \log P(x) - D_{KL}(Q(z)||P(z|x)). \qquad (45)$$

We began by comparing massively parallel and global importance sampling (Fig. 1) on each dataset using the predefined (and fixed) generative and proposal distributions. We found that for a given value of $K$, the ELBO (top) and predictive log-likelihood (bottom) is far better for massively parallel than global importance sampling, as massively parallel sampling considers all $K^n$ combinations of all samples of all latent variables, while global importance sampling considers only $K$ samples from the full joint latent space.

However, for a fixed $K$, the time taken for massively parallel importance sampling is much larger than the time taken for global importance sampling (and this also meant that we used a far smaller range of $K$ for massively parallel importance sampling). We therefore also plotted the ELBO (top) and predictive log-likelihood (bottom) with time on the x-axis (Fig. 2). This again shows that massively parallel importance sampling gives large improvements in ELBO and predictive log-likelihood for a fixed time budget.

Finally, we compared massively parallel importance sampling against iterative methods such as VI, IWAE and

RWS that learn a better proposal (Fig. 3). These methods use the factorised proposal distribution $Q\left(z^{\mathbf{k}}\right)$ as their initial approximate posterior and iteratively update the parameters of each latent variable's proposal distribution (in most cases this is the mean and variance of a Gaussian—see Appendix section C for full details). VI and IWAE update the proposal parameters in order to maximise the global ELBO, the former with $K = 1$ and the latter with $K > 1$ (in particular we used $K = 10$). RWS performs a maximum-likelihood update on the parameters of $Q$ using posterior samples obtained by reweighting the proposal samples with the importance weights $r_k(z)$ given in Eq. 6, also with $K = 10$. In each case, this update is performed using the Adam optimizer with default hyperparameters and tuned learning rates which are discussed in Appendix sections C.5 and D.

Of course, learning a good proposal is critical to the effectiveness of practical inference methods. Thus, comparing massively parallel importance sampling against these methods is not really fair, as massively parallel importance sampling is a "one-shot" method that is forced to use an extremely poor proposal (the prior). To do a fair comparison, we would need to build massively parallel importance sampling into an iterative method that iteratively improved the proposal (but this is a considerable endeavour which is out of scope for the present work). Even with these fundamental limitations, massively parallel importance sampling seems to fill out an important part of the speed-accuracy tradeoff. In particular, massively parallel importance sampling gives us good results extremely rapidly, while iterative methods take at least an order of magnitude longer to reach similar ELBOs and predictive log-

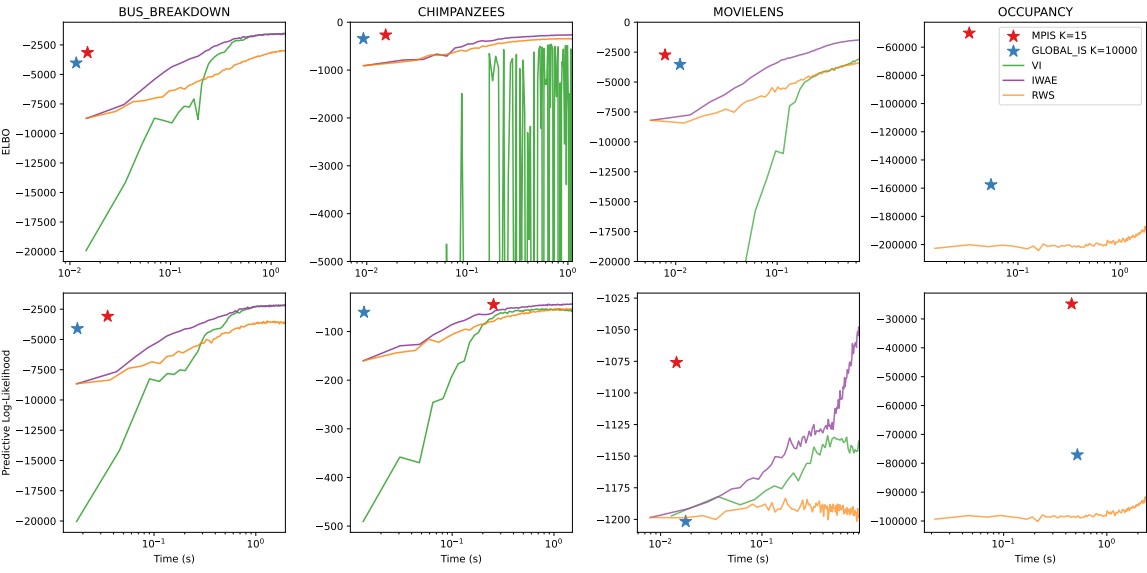

Figure 3: Comparing the ELBO and predicitve log-likelihood of MP IS with K=15 (red star), global IS with $K = 10,000$ (blue star), VI (green lines), IWAE (purple lines) and RWS (orange lines). For MP IS and global IS we have plotted the highest K value that we were able to compute for all models, specifically $K = 15$ and $K = 10,000$ respectively. Single-sample VI was performed (i.e. $K = 1$), whereas IWAE and RWS were done using $K = 10$. The only iterative method we report for the Occupancy model is RWS, as this model contains discrete latent variables which precludes the use of gradient-based methods such as VI and IWAE.

likelihoods (Fig. 3). Here, we plotted a single line for each method (VI, IWAE and RWS), corresponding to the best learning rate (see Appendix D). We considered including HMC in Fig. 3, but it turned out not to be possible because HMC methods took longer to return even a single sample than the time plotted. These long timescales arise because HMC requires many gradient evaluations for a single HMC step, and these steps are inherently sequential and therefore cannot make good use of GPU parallelism. And this is even before we consider the need for adaptation and burn-in necessary in all MCMC methods.

## 5 RELATED WORK

There is a considerable body of work in the discrete graphical model setting that computes posterior expectations, marginals and samples [Dawid, 1992, Pfeffer, 2005, Bidyuk and Dechter, 2007, Geldenhuys et al., 2012, Gogate and Dechter, 2012, Claret et al., 2013, Sankaranarayanan et al., 2013, Goodman and Stuhlmüller, 2014, Gehr et al., 2016, Narayanan et al., 2016, Albarghouthi et al., 2017, Dechter et al., 2018, Wang et al., 2018, Obermeyer et al., 2019, Holtzen et al., 2020]. Our work differs in two respects. First, our massively parallel methods are not restricted to discrete graphical models, but can operate with arbitrary continuous latent variables and graphs with a mixture of continuous and discrete latent variables. Second, this prior work involves

complex implementations that, in one sense or another, "proceed by recording an adjoint compute graph alongside the forward computation and then traversing the adjoint graph backwards starting from the final result of the forward computation" [Obermeyer et al., 2019]. The forward computation is reasonably straightforward: it is just a big tensor product that can be computed efficiently using pre-existing libraries such as opt-einsum, and results in (an estimate of) the marginal likelihood. However, the backward traversal is much more complex, if for no other reason than the need to implement separate traversals for each operation of interest (computing posterior expectations, marginals and samples). Additionally, these traversals need to correctly handle all special cases, including optimized implementations of plates and timeseries. Importantly, optimizing the forward computation is usually quite straightforward while implementing an optimized backward traversal is far more complex. For instance, the forward computation for a timeseries involves a product of $T$ matrices arranged in a chain. Naively computing this product on GPUs is very slow, as it requires $T$ separate matrix multiplications. However, it is possible to massively optimize this forward computation, converting $\mathcal{O}(T)$ to $\mathcal{O}(\log(T))$ tensor operations by multiplying adjacent pairs of matrices in a single batched matrix multiplication operation. This optimization is straightforward in the forward computation. However, applying this optimization as part of the backward computation is far more complex (see Corenflos et al., 2022 for details). This complexity,

along with similar complexity for other important optimizations such as plates, is prohibitive for academic teams implementing e.g. new probabilistic programming languages. Our key contribution is thus to provide a much simpler approach to directly compute posterior expectations, marginals and samples by differentiating through the forward computation, without having to hand-write and hand-optimize backward traversals.

There is work on fitting importance weighted autoencoders [IWAE; Burda et al., 2015] and reweighted wake-sleep [RWS; Bornschein and Bengio, 2014, Le et al., 2020] in the massively parallel setting [Aitchison, 2019, Geffner and Domke, 2022, Heap et al., 2023] for general probabilistic models. However, this work only provides methods for performing massively parallel updates to approximate posteriors (e.g. by optimizing a massively parallel ELBO). This work does not provide a method to individually reweight the samples to provide accurate posterior expectations, marginals and samples. Instead, this previous work simply takes the learned approximate posterior as an estimate of the true posterior, and does not attempt to correct for inevitable differences between approximate and true posterior.

Massively parallel importance sampling itself, bears similarities to e.g. particle filtering/SMC methods [Gordon et al., 1993, Doucet et al., 2009, Andrieu et al., 2010, Maddison et al., 2017, Le et al., 2017, Lindsten et al., 2017, Naesseth et al., 2018, Kuntz et al., 2023, Lai et al., 2022, Crucinio and Johansen, 2023] that have been generalised to arbitrary graphical models and where the resampling step has been eliminated. However, our contribution is not massively parallel importance sampling in itself. Instead, our contribution is the simple method, using autodiff to differentiate through a marginal likelihood estimator, for computing posterior expectations, marginals and samples without requiring the implementation of complex backwards traversals, and this has not appeared in past work.

# 6 CONCLUSION

We gave a new and far simpler method for computing posterior moments, marginals and samples in massively parallel importance sampling based on differentiating a slightly modified marginal likelihood estimator.

The method has limitations, in that while it is considerably more effective than e.g. VI, RWS and global importance sampling, it is more complex. Additionally, at least a naive implementation may be quite costly in terms of memory consumption, limiting how the number of importance samples we can draw for each variable. That said, it should be possible to eliminate almost all of this overhead by careful optimizations to avoid allocating large intermediate tensors, following the strategy in KeOps [Charlier et al., 2021].

For future work, we intend to use the contributions from

this paper in developing an iterative inference algorithm, similar to VI and RWS, that uses massively parallel importance sampling, as well as developing a massively parallel probabilistic programming language.

# Acknowledgements

Sam Bowyer is supported by the UKRI Engineering and Physical Sciences Research Council via the COMPASS Centre for Doctoral Training at the University of Bristol (EP/S023569/1). Thomas Heap is also supported by the UKRI Engineering and Physical Sciences Research Council.

This work was possible thanks to the computational facilities of the University of Bristol's Advanced Computing Research Centre—http://www.bris.ac.uk/acrc/. We would like to thank Dr. Stewart for funding compute resources used in this project.

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

# Using Autodiff to Estimate Posterior Moments, Marginals and Samples (Supplementary Material)

**Sam Bowyer**[1]          **Thomas Heap**[2]          **Laurence Aitchison**[2]

[1]School of Mathematics, University of Bristol
[2]Department of Computer Science, University of Bristol

## A  DERIVATIONS

### A.1  GLOBAL IMPORTANCE SAMPLING

Here, we give the derivation for standard global importance sampling. Ideally we would compute moments using the true posterior, $\mathrm{P}\left(z|x\right)$,

$$m_{\text{post}} = \mathrm{E}_{\mathrm{P}\left(z^k|x\right)}\left[m(z^k)\right]. \tag{46}$$

However, the true posterior is not known. Instead, we write down the moment under the true posterior as an integral,

$$m_{\text{post}} = \int dz^k\,\mathrm{P}\left(z^k|x\right)m(z^k). \tag{47}$$

Next, we multiply the integrand by $1 = \mathrm{Q}\left(z^k\right)/\mathrm{Q}\left(z^k\right)$,

$$m_{\text{post}} = \int dz^k \mathrm{Q}\left(z^k\right)\frac{\mathrm{P}\left(z^k|x\right)}{\mathrm{Q}\left(z^k\right)}m(z^k). \tag{48}$$

Next, the integral can be written as an expectation,

$$m_{\text{post}} = \mathrm{E}_{\mathrm{Q}(z^k)}\left[\frac{\mathrm{P}\left(z^k|x\right)}{\mathrm{Q}\left(z^k\right)}m(z^k)\right]. \tag{49}$$

It looks like we should be able to estimate $m_{\text{post}}$ by sampling from our approximate posterior, $\mathrm{Q}\left(z^k\right)$. However, this is not yet possible, as we are not able to compute the true posterior, $\mathrm{P}\left(z^k|x\right)$. We might consider using Bayes theorem,

$$\mathrm{P}\left(z^k|x\right) = \frac{\mathrm{P}\left(z^k,x\right)}{\mathrm{P}\left(x\right)}, \tag{50}$$

but this requires computing an intractable normalizing constant,

$$\mathrm{P}\left(x\right) = \int dz^k\,\mathrm{P}\left(z^k,x\right). \tag{51}$$

Instead, we use an unbiased, importance-sampled estimate of the normalizing constant, $\mathcal{P}_{\text{global}}(z)$ (Eq. 7). Burda et al. [2015] showed that in the limit as $K \to \infty$, $\mathcal{P}_{\text{global}}(z)$ approaches $\mathrm{P}\left(x\right)$. Using this estimate of the marginal likelihood our moment estimate becomes,

$$m_{\text{global}} = \mathrm{E}_{\mathrm{Q}(z^k)}\left[\frac{\frac{\mathrm{P}\left(z^k,x\right)}{\mathrm{Q}(z^k)}}{\mathcal{P}_{\text{global}}(z)}m(z^k)\right]. \tag{52}$$

Using $r_k(z)$ (Eq. 6), we can write this expression as,

$$m_{\text{global}} = \mathrm{E}_{\mathrm{Q}_\phi(z)}\left[\frac{r_k(z)}{\mathcal{P}_{\text{global}}(z)}m(z^k)\right].$$

(53)

The approximate posterior and generative probabilities are the same for different values of $k$, so we can average over $k$, which gives Eq. (5) in the main text.

## A.2 MASSIVELY PARALLEL IMPORTANCE SAMPLING

Inspired by the global importance sampling derivation, we consider massively parallel importance sampling. In the global importance sampling derivation, the key idea was to show that the estimator was unbiased for each of the $K$ samples, $z^k$, in which case the average over all $K$ samples is also unbiased. In massively parallel importance sampling, we use the same idea, except that we now have $K^n$ samples, denoted $z^{\mathbf{k}}$. As before,

$$m_{\text{post}} = \mathrm{E}_{\mathrm{P}(z^{\mathbf{k}}|x)}\left[m(z^{\mathbf{k}})\right] = \int dz^{\mathbf{k}}\,\mathrm{P}\left(z^{\mathbf{k}}|x\right)m(z^{\mathbf{k}}).$$

(54)

Again, we multiply and divide by the approximate posterior for $z^{\mathbf{k}}$. In the massively parallel setting, we specifically use $\prod_i \mathrm{Q}\left(z_i^{k_i}\middle|z_{\text{qa}(i)}\right)$,

$$m_{\text{post}} = \int dz^{\mathbf{k}}\left(\prod_i \mathrm{Q}\left(z_i^{k_i}\middle|z_{\text{qa}(i)}\right)\right)\frac{\mathrm{P}\left(z^{\mathbf{k}}|x\right)}{\prod_i \mathrm{Q}\left(z_i^{k_i}\middle|z_{\text{qa}(i)}\right)}m(z^{\mathbf{k}}).$$

(55)

Overall, our goal is to convert the integral over the indexed latent variables in Eq. (55) into an integral over the full latent space, $z$, so that it can be written as an expectation over the proposal, $\mathrm{Q}(z)$. To do that, we need to introduce the concept of non-indexed latent variables. These are all samples of the latent variables, except for the "indexed", or $k$th sample. For the $i$th latent variable, the non-indexed samples are,

$$z_i^{/k_i} = \left(z_i^1, \ldots, z_i^{k_i-1}, z_i^{k_i+1}, \ldots, z_i^K\right) \in \mathcal{Z}_i^{K-1}.$$

(56)

We can also succinctly write the non-indexed samples of all latent variables as,

$$z^{/\mathbf{k}} = \left(z_1^{/k_1}, z_2^{/k_2}, \ldots, z_n^{/k_n}\right) \in \mathcal{Z}^{K-1}.$$

(57)

The joint distribution over the non-indexed latent variables, conditioned on the indexed latent variables integrates to 1,

$$1 = \int dz^{/\mathbf{k}}\prod_i \mathrm{Q}\left(z_i^{/k_i}\middle|z_i^{k_i}, z_{\text{qa}(i)}\right),$$

(58)

We use this to multiply the integrand in Eq. (55),

$$m_{\text{post}} = \int dz^{\mathbf{k}}\left(\prod_i \mathrm{Q}\left(z_i^{k_i}\middle|z_{\text{qa}(i)}\right)\right)\frac{\mathrm{P}\left(z^{\mathbf{k}}|x\right)}{\prod_i \mathrm{Q}\left(z_i^{k_i}\middle|z_{\text{qa}(i)}\right)}m(z^{\mathbf{k}})\int dz^{/\mathbf{k}}\prod_i \mathrm{Q}\left(z_i^{/k_i}\middle|z_i^{k_i}, z_{\text{qa}(i)}\right).$$

(59)

Next, we merge the integrals over over $z^{\mathbf{k}}$ and $z^{/\mathbf{k}}$ to form one integral over $z$,

$$m_{\text{post}} = \int dz\,\mathrm{Q}(z)\frac{\mathrm{P}\left(z^{\mathbf{k}}|x\right)}{\prod_i \mathrm{Q}\left(z_i^{k_i}\middle|z_{\text{qa}(i)}\right)}m(z^{\mathbf{k}}).$$

(60)

This integral can be written as an expectation,

$$m_{\text{post}} = \mathrm{E}_{\mathrm{Q}(z)}\left[\frac{\mathrm{P}\left(z^{\mathbf{k}}|x\right)}{\prod_i \mathrm{Q}\left(z_i^{k_i}|z_{\text{pa}(i)}\right)}m(z^{\mathbf{k}})\right].$$

(61)

---
**Algorithm 1** Expectations using the source term trick

---

**Require:** Data $x$, Generative Model $P(\mathbf{x}|z)$, Prior $P(\mathbf{z})$, Proposal $Q_{\text{MP}}$, $K \geq 1$, function whose expectation we want to compute $m(z^{\mathbf{k}})$

    **for** $i \leftarrow 1$ to $n$ **do**

        Sample $z_i \sim Q_{\text{MP}}\left(z_i | z_j \text{ for all } j \in \text{qa}(i)\right)$

        $z \leftarrow \{z_1, ..., z_{i-1}\} \cup z_i$

        $f^i_{k_i, \mathbf{k}_{\text{pa}(i)}}(z) \leftarrow \dfrac{P\left(z_i^{k_i} \big| z_j^{k_j} \text{ for all } j \in \text{pa}(i)\right)}{Q_{\text{MP}}\left(z_i^{k_i} | x, z_j \text{ for all } j \in \text{qa}(i)\right)}$

    **end for**

    $f^x_{\mathbf{k}_{\text{pa}(x)}}(z) \leftarrow P\left(x \big| z_j^{k_j} \text{ for all } j \in \text{pa}(x)\right)$

    $r_{\mathbf{k}}(z) \leftarrow f^x_{\mathbf{k}_{\text{pa}(x)}}(z) \prod_i f^i_{k_i, \mathbf{k}_{\text{pa}(i)}}(z)$

    $J \leftarrow 0$

    $\mathcal{P}^{\text{exp}}_{\text{MP}}(z, J) \leftarrow \frac{1}{K^n} \sum_{\mathbf{k}^n} r_{\mathbf{k}}(z) e^{J m(z^{\mathbf{k}})}$

    **return** $m_{\text{MP}}(z) \leftarrow \frac{\partial}{\partial J}\big|_{J=0} \log \mathcal{P}^{\text{exp}}_{\text{MP}}(z, J)$

---

As in the derivation for global importance sampling, it looks like we might be able to estimate this by sampling from $Q(z|x)$, but this does not yet work as we do not yet have a form for the posterior. Again, we could compute the posterior using Bayes theorem,

$$P\left(z^{\mathbf{k}}|x\right) = \frac{P\left(z^{\mathbf{k}}, x\right)}{P(x)}, \tag{62}$$

but we cannot compute the model evidence,

$$P_\theta(x) = \int dz^{\mathbf{k}} P_\theta\left(z^{\mathbf{k}}, x\right). \tag{63}$$

As in the global importance sampling section, we instead use an estimate of the marginal likelihood. Here, we use a massively parallel estimate, $\mathcal{P}_{\text{MP}}(z)$,

$$m_{\text{MP}} = E_{Q(z|x)}\left[\frac{\frac{P\left(z^{\mathbf{k}}, x\right)}{\prod_i Q\left(z_i^{k_i} | z_{\text{pa}(i)}\right)}}{\mathcal{P}_{\text{MP}}(z)} m(z^{\mathbf{k}})\right]. \tag{64}$$

Again, we use $r_{\mathbf{k}}(z)$ (Eq. 18),

$$m_{\text{MP}} = E_{Q(z)}\left[\frac{r_{\mathbf{k}}(z)}{\mathcal{P}_{\text{MP}}(z)} m(z^{\mathbf{k}})\right]. \tag{65}$$

So the value for a single set of latent variables, $z^{\mathbf{k}}$, has the right expectation. Thus, averaging over all $K^n$ settings of $\mathbf{k}$, we get the unbiased estimator in the main text, (Eq. 24).

## B   ALGORITHMS

---
**Algorithm 2** Marginal importance weights using the source term trick
---
**Require:** Data $x$, Generative Model $\mathrm{P}(\mathbf{x}|z)$, Prior $\mathrm{P}(\mathbf{z})$, Proposal $\mathrm{Q}_{\mathrm{MP}}$, $K \geq 1$, index of latent for which to calculate weight: $i$

    **for** $i \leftarrow 1$ to $n$ **do**

        Sample $z_i \sim \mathrm{Q}_{\mathrm{MP}}\left(z_i | z_j \text{ for all } j \in \mathrm{qa}\,(i)\right)$

        $z \leftarrow \{z_1, ..., z_{i-1}\} \cup z_i$

        $f^i_{k_i, \mathbf{k}_{\mathrm{pa}(i)}}(z) \leftarrow \dfrac{\mathrm{P}\left(z_i^{k_i} \middle| z_j^{k_j} \text{ for all } j \in \mathrm{pa}(i)\right)}{\mathrm{Q}_{\mathrm{MP}}\left(z_i^{k_i} | x, z_j \text{ for all } j \in \mathrm{qa}(i)\right)}$

    **end for**

    $f^x_{\mathbf{k}_{\mathrm{pa}(x)}}(z) \leftarrow \mathrm{P}\left(x \middle| z_j^{k_j} \text{ for all } j \in \mathrm{pa}(x)\right)$

    $r_\mathbf{k}(z) \leftarrow f^x_{\mathbf{k}_{\mathrm{pa}(x)}}(z) \prod_i f^i_{k_i, \mathbf{k}_{\mathrm{pa}(i)}}(z)$

    $\mathbf{J} \leftarrow \mathbf{0} \in \mathbb{R}^K$

    $\mathcal{P}^{\mathrm{marg}}_{\mathrm{MP}}(z, \mathbf{J}) \leftarrow \frac{1}{K^n} \sum_{\mathbf{k}^n} r_\mathbf{k}(z) e^{J_{k_i}}$

    **return** $w^i_{k_i} \leftarrow \left.\frac{\partial}{\partial J_{k_i}}\right|_{J=0} \log \mathcal{P}^{\mathrm{marg}}_{\mathrm{MP}}(z, \mathbf{J})$

---

---
**Algorithm 3** Joint distributions using the source term trick
---
**Require:** Data $x$, Generative Model $\mathrm{P}(\mathbf{x}|z)$, Prior $\mathrm{P}(\mathbf{z})$, Proposal $\mathrm{Q}_{\mathrm{MP}}$, $K \geq 1$, index of latent for which to compute conditional distribution: $i$

    **for** $i \leftarrow 1$ to $n$ **do**

        Sample $z_i \sim \mathrm{Q}_{\mathrm{MP}}\left(z_i | z_j \text{ for all } j \in \mathrm{qa}\,(i)\right)$

        $z \leftarrow \{z_1, ..., z_{i-1}\} \cup z_i$

        $f^i_{k_i, \mathbf{k}_{\mathrm{pa}(i)}}(z) \leftarrow \dfrac{\mathrm{P}\left(z_i^{k_i} \middle| z_j^{k_j} \text{ for all } j \in \mathrm{pa}(i)\right)}{\mathrm{Q}_{\mathrm{MP}}\left(z_i^{k_i} | x, z_j \text{ for all } j \in \mathrm{qa}(i)\right)}$

    **end for**

    $f^x_{\mathbf{k}_{\mathrm{pa}(x)}}(z) \leftarrow \mathrm{P}\left(x \middle| z_j^{k_j} \text{ for all } j \in \mathrm{pa}(x)\right)$

    $\mathbf{J} \leftarrow \mathbf{0} \in \mathbb{R}^{K^{1+|\mathrm{pa}(i)|}}$

    $r_\mathbf{k}(z) \leftarrow f^x_{\mathbf{k}_{\mathrm{pa}(x)}}(z) \prod_i f^i_{k_i, \mathbf{k}_{\mathrm{pa}(i)}}(z)$

    $\mathcal{P}^{\mathrm{samp}}_{\mathrm{MP}}(z, \mathbf{J}) \leftarrow \frac{1}{K^n} \sum_{\mathbf{k}} \frac{\mathrm{P}\left(x, z^\mathbf{k}\right)}{\mathrm{Q}(z^\mathbf{k})} e^{J_{k_i, \mathbf{k}_{\mathrm{pa}(i)}}}$

    **return** $\mathrm{P}\left(k_i, \mathbf{k}_{\mathrm{pa}(i)}\right) \leftarrow \left.\frac{\partial}{\partial J_{k_i, \mathbf{k}_{\mathrm{pa}(i)}}}\right|_{\mathbf{J}=\mathbf{0}} \log \mathcal{P}^{\mathrm{samp}}_{\mathrm{MP}}(z, \mathbf{J})$

---

# C EXPERIMENTAL DATASETS AND MODELS

## C.1 BUS DELAY DATASET

In this experiment, we model the length of delays to New York school bus journeys,[1] working with a dataset supplied by the City of New York [DOE, 2023]. Our goal is to predict the length of a delay, based on the year, $y$, borough, $b$, bus company, $c$ and journey type, $j$. Specifically, our data includes the years $2015 - 2022$ inclusive and covers the five New York boroughs (Brooklyn, Manhatten, The Bronx, Queens, Staten Island) as well as some surrounding areas (Nassau County, New Jersey, Connecticut, Rockland County, Westchester). There are 57 bus companies, and 6 different journey types (e.g. pre-K/elementary school route, general education AM/PM route etc.) We take $I = 60$ delayed buses in each borough and year, and take $Y = 3$ years and $B = 3$ boroughs. We then split along the $I$ dimension to get two equally sized train and test sets. Thus, each delay is uniquely identified by the year, $y$, the borough, $b$, and the index, $i$, giving $\mathrm{delay}_{ybi}$ The delays are recorded as an integer number of minutes and we discard any entries greater than 130 minutes.

We have a hierarchical latent variable model describing the impact of each of the features (year, borough, bus company and journey type) on the length of delays. Specifically, the integer delay is modelled with a Negative Binomial distribution, with fixed total count of 131. The expected delay length is controlled by a logits latent variable, $\mathrm{logits}_{ybi}$, with one logits for each

---

[1]Dataset: `data.cityofnewyork.us/Transportation/Bus-Breakdown-and-Delays/ez4e-fazm`
Terms of use: `//opendata.cityofnewyork.us/overview/#termsofuse`

delayed bus. The logits is a sum of three terms: one for the borough and year jointly, one for the bus company and one for the journey type. Each of these three terms is themselves a latent variable that must be inferred.

First, we have a term for the year and borough, $\text{YearBoroughWeight}_{yb}$, which has a hierarchical prior. Specifically, we begin by sampling a global mean and variance, GlobalMean and GlobalVariance. Then for each year, we use GlobalMean and GlobalVariance to sample a mean for each year, $\text{YearMean}_y$. Additionally, we sample a variance for each year, $\text{YearVariance}_y$. Then we sample a $\text{YearBoroughWeight}_{yb}$ from a Gaussian distribution with a year-dependent mean, $\text{YearMean}_y$, and variance $\exp(\text{BoroughVariance}_b)$.

Next, the weights for the bus company and journey type are very similar. Specifically, we have one latent weight for each bus company, $\text{CompanyWeight}_c$, with $c \in \{1, \ldots, 57\}$, and for each journey type, $\text{JourneyTypeWeight}_j$, with $j \in \{1, \ldots, 6\}$. We have a table identifying the bus company, $b_{ybi}$, and journey type, $j_{ybi}$, for each delayed bus journey (remember that a particular delayed bus journey is uniquely identified by the year, $y$, borough, $b$ and index $i$). In $\text{logits}_{ybi}$ we use these tables to pick out the right company and journey type weight for that particular delayed bus journey, $\text{CompanyWeight}_{c_{ybi}}$ and $\text{JourneyTypeWeight}_{j_{ybi}}$. The final generative model is defined by

$$
\begin{aligned}
\mathrm{P}\left(\text{GlobalVariance}\right) &= \mathcal{N}(\text{GlobalVariance}; 0, 1) \\
\mathrm{P}\left(\text{GlobalMean}\right) &= \mathcal{N}(\text{GlobalMean}; 0, 1) \\
\mathrm{P}\left(\text{YearMean}_y | \text{GlobalMean}, \text{GlobalVariance}\right) &= \mathcal{N}(\text{YearMean}_y; \text{GlobalMean}, \exp(\text{GlobalVariance})), \\
& \quad y \in \{1, ..., Y\} \\
\mathrm{P}\left(\text{YearVariance}_b\right) &= \mathcal{N}(\text{YearVariance}_b; 0, 1), \\
& \quad y \in \{1, ..., Y\} \\
\mathrm{P}\left(\text{YearBoroughWeight}_{yb} | \text{YearMean}_y, \text{YearVariance}_b\right) &= \mathcal{N}(\text{YearBoroughWeight}_{yb}; \text{YearMean}_y, \exp(\text{YearVariance}_b)), \\
& \quad b \in \{1, ..., B\} \\
\mathrm{P}\left(\text{CompanyWeight}_c\right) &= \mathcal{N}(\text{CompanyWeight}_c; 0, 1), \ c \in \{1, ..., C\} \\
\mathrm{P}\left(\text{JourneyTypeWeight}_j\right) &= \mathcal{N}(\text{JourneyTypeWeight}_j; 0, 1), \ j \in \{1, ..., J\} \\
\text{logits}_{ybi} &= \text{YearBoroughWeight}_{yb} + \text{CompanyWeight}_{c_{ybi}} \\
& \quad + \text{JourneyTypeWeight}_{j_{ybi}} \\
\mathrm{P}\left(\text{delay}_{ybi} | \text{logits}_{ybi}\right) &= \text{NegativeBinomial}(\text{delay}_{ybi}; \text{total count} = 131, \text{logits}_{ybi}),
\end{aligned}
$$
(66)

and the corresponding graphical model is given in Fig. 4. We define the factorised proposal distribution $Q$ (which acts also as the initial approximate posterior for the iterative methods in Fig. 3) in a very similar fashion:

$$
\begin{aligned}
\mathrm{Q}\left(\text{GlobalVariance}\right) &= \mathcal{N}(\text{GlobalVariance}; 0, 1) \\
\mathrm{Q}\left(\text{GlobalMean}\right) &= \mathcal{N}(\text{GlobalMean}; 0, 1) \\
\mathrm{Q}\left(\text{YearMean}_y\right) &= \mathcal{N}(\text{YearMean}_y; 0, 1), \ y \in \{1, ..., Y\} \\
\mathrm{Q}\left(\text{YearVariance}_b\right) &= \mathcal{N}(\text{YearVariance}_b; 0, 1), \ y \in \{1, ..., Y\} \\
\mathrm{Q}\left(\text{YearBoroughWeight}_{yb}\right) &= \mathcal{N}(\text{YearBoroughWeight}_{yb}; 0, 1), \ b \in \{1, ..., B\} \\
\mathrm{Q}\left(\text{CompanyWeight}_c\right) &= \mathcal{N}(\text{CompanyWeight}_c; 0, 1), \ c \in \{1, ..., C\} \\
\mathrm{Q}\left(\text{JourneyTypeWeight}_j\right) &= \mathcal{N}(\text{JourneyTypeWeight}_j; 0, 1), \ j \in \{1, ..., J\}
\end{aligned}
$$
(67)

## C.2 CHIMPANZEE PROSOCIALITY DATASET

The Chimpanzee Prosociality dataset[2] consists of repeated experiments in a controlled setting for testing the prosociable tendencies of seven chimpanzees [Silk et al., 2005]. In each experiment, the chimpanzee being tested, the focal chimpanzee,

---
[2]Dataset: `https://rdrr.io/github/rmcelreath/rethinking/man/chimpanzees.html`
License (GPL V3): `https://github.com/rmcelreath/rethinking`

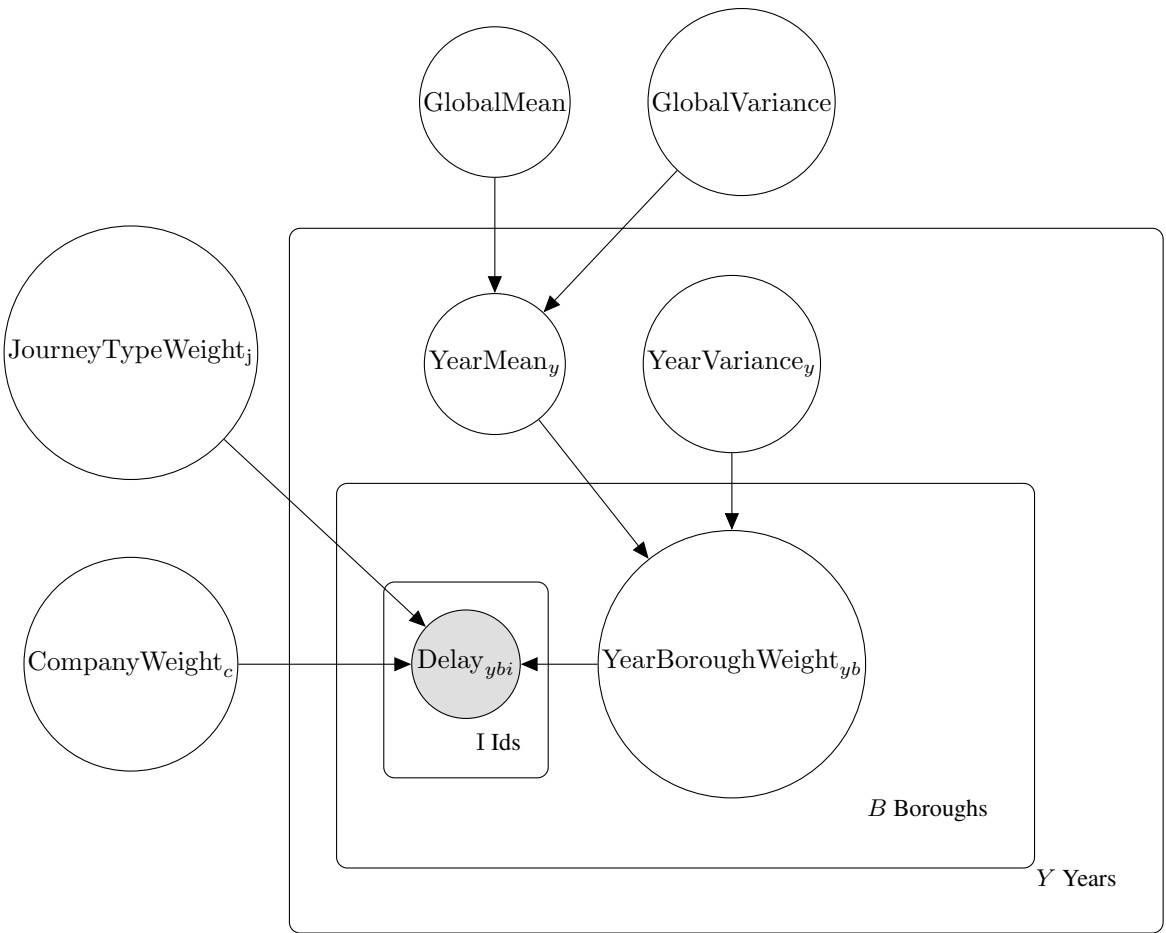

Figure 4: Graphical model for the NYC Bus Breakdown dataset

would be sat at the end of a table with two levers in front of them, one on their left and one on their right. Each lever was connected to two dishes on the corresponding left or right side of the table, one closer to the focal chimpanzee than the other. Pulling the lever would slide those two dishes to opposite ends of the table, one towards the focal chimpanzee and one the other away. Every repeat of the experiment, the two dishes closest to the chimpanzee (one on the right and one on the left) contained food, however, only one of the two dishes towards the end of the table contained any food. This meant that the focal chimpanzee received food no matter which lever was pulled, but had the option of whether to send food to the other end of the table or not. This experiment was repeated $R = 12$ times, on $B = 6$ separate occasions ("blocks") per $A = 7$ chimpanzees ("actors"), each time potentially with another chimpanzee sat at the opposite of the table (with no levers), and with the empty dish sometimes connected to the left lever and sometimes connected to the right lever.

We base our hierarchical model on the cross-classified varying slopes model presented in McElreath [2016], where the observations $y_{abr}$ represent whether a given chimpanzee $a \in \{1, ..., A\}$ during a repeat $r \in \{1, ..., R\}$ inside a particular block $b \in \{1, ..., B\}$ of experiments pulled the left lever. To help identify prosociable behaviour we work with two binary covariate quantities for a given combination $(a, b, r)$: $\text{Condition}_{abr} \in \{0, 1\}$ indicating whether or not another chimpanzee sat at the other end of the table; and $\text{ProsocLeft}_{abr}$ indicating whether the prosociable lever (connected to two full dishes, rather than just one) was on the left or not. We model the observations via a Bernoulli variable whose logits are given by the sum of two quantities: an intercept model $\alpha + \alpha_a + \alpha_{ab}$ and a slope model $(\beta_{\text{P}} + \beta_{\text{PC}} * \text{Condition}_{abr}) * \text{ProsocLeft}_{abr}$.

The intercept model is comprised of three quantities: a global intercept $\alpha$ with a zero-mean Normal prior with variance 10, a per-actor intercept $\alpha_a$ and a per-actor-block intercept $\alpha_{ab}$, the latter of which have zero-mean priors and each have variances, $\sigma_{\text{ACTOR}^2}, \sigma_{\text{BLOCK}^2}$ respectively sampled from a hyperprior $\text{HalfCauchy}(1)$ distribution. The slope model is comprised of two latent variables, $\beta_{\text{PC}}, \beta_{\text{P}}$, representing the effect on the chimpanzee of the presence of another chimpanzee at the end of the table and whether the prosocial choice came from the left lever or not respectively. Both of these variables are sampled

from a Normal distribution with mean zero and variance 10. The full specification of the generative model is given by

$$
\begin{aligned}
\mathrm{P}\left(\sigma_{\mathrm{ACTOR}}^2\right) &= \mathrm{HalfCauchy}(\sigma_{\mathrm{ACTOR}}^2; 1), \\
\mathrm{P}\left(\sigma_{\mathrm{BLOCK}}^2\right) &= \mathrm{HalfCauchy}(\sigma_{\mathrm{BLOCK}}^2; 1), \\
\mathrm{P}\left(\beta_{\mathrm{PC}}\right) &= \mathcal{N}(\beta_{\mathrm{PC}}; 0, 10), \\
\mathrm{P}\left(\beta_{\mathrm{P}}\right) &= \mathcal{N}(\beta_{\mathrm{P}}; 0, 10), \\
\mathrm{P}\left(\alpha\right) &= \mathcal{N}(\alpha; 0, 10), \\
\mathrm{P}\left(\alpha_a | \sigma_{\mathrm{ACTOR}}^2\right) &= \mathcal{N}(\alpha_a; 0, \sigma_{\mathrm{ACTOR}}^2), \quad a \in \{1, ..., A\} \\
\mathrm{P}\left(\alpha_{ab} | \sigma_{\mathrm{BLOCK}}^2\right) &= \mathcal{N}((\alpha_{ab}; 0, \sigma_{\mathrm{BLOCK}}^2), \quad b \in \{1, ..., B\} \\
\mathrm{logits}_{abr} &= \alpha + \alpha_a + \alpha_{ab} + (\beta_{\mathrm{P}} + \beta_{\mathrm{PC}} * \mathrm{Condition}_{abr}) * \mathrm{ProsocLeft}_{abr}, \quad r \in \{1, ..., R\} \\
\mathrm{P}\left(y_{abr} | \mathrm{logits}_{abr}\right) &= \mathrm{Bernoulli}(y_{abr}; \mathrm{logits} = \mathrm{logits}_{abr})
\end{aligned}
\tag{68}
$$

and the graphical model is given in Fig. 5.

The factorised proposal distribution $Q$ is defined similarly:

$$
\begin{aligned}
\mathrm{Q}\left(\sigma_{\mathrm{ACTOR}}^2\right) &= \mathrm{HalfCauchy}(\sigma_{\mathrm{ACTOR}}^2; 1), \\
\mathrm{Q}\left(\sigma_{\mathrm{BLOCK}}^2\right) &= \mathrm{HalfCauchy}(\sigma_{\mathrm{BLOCK}}^2; 1), \\
\mathrm{Q}\left(\beta_{\mathrm{PC}}\right) &= \mathcal{N}(\beta_{\mathrm{PC}}; 0, 10), \\
\mathrm{Q}\left(\beta_{\mathrm{P}}\right) &= \mathcal{N}(\beta_{\mathrm{P}}; 0, 10), \\
\mathrm{Q}\left(\alpha\right) &= \mathcal{N}(\alpha; 0, 10), \\
\mathrm{Q}\left(\alpha_a\right) &= \mathcal{N}(\alpha_a; 0, 1), \quad a \in \{1, ..., A\} \\
\mathrm{Q}\left(\alpha_{ab}\right) &= \mathcal{N}(\alpha_{ab}; 0, 1), \quad b \in \{1, ..., B\}
\end{aligned}
\tag{69}
$$

In our experiments, we split the data into a training set that takes $R = 10$ of the repeats for each actor-block combination, and a test set that takes the remaining $R = 2$.

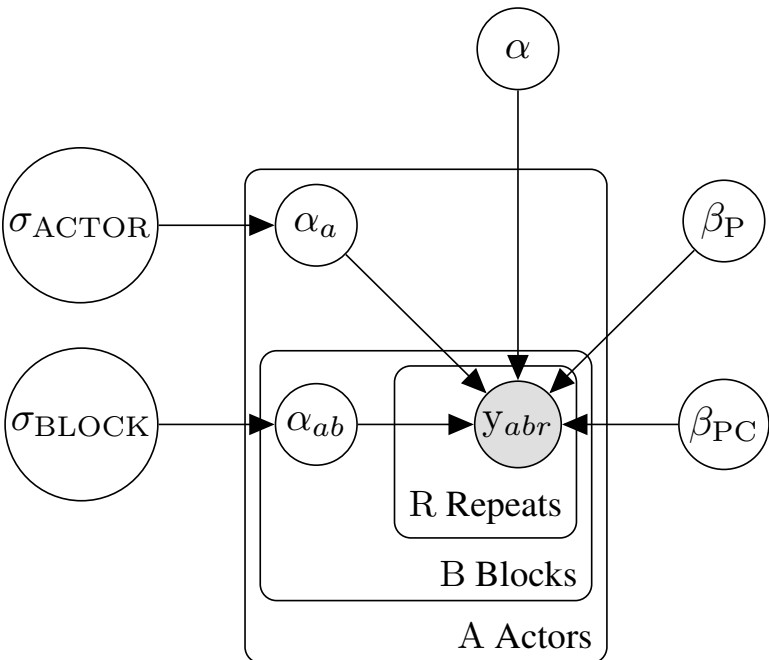

Figure 5: Graphical model for the Chimpanzees dataset

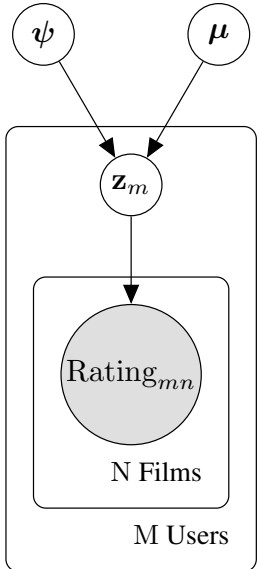

Figure 6: Graphical model for the MovieLens dataset

### C.3 MOVIELENS DATASET

The MovieLens100K[3] [Harper and Konstan, 2015] dataset contains 100k ratings of $N{=}1682$ films from among $M{=}943$ users. The original user ratings run from 0 to 5. Following Geffner and Domke [2022], we binarise the ratings into just likes/dislikes, by taking user-ratings of $\{0, 1, 2, 3\}$ as a binarised-rating of 0 (dislikes) and user-ratings of $\{4, 5\}$ as a binarised-rating of 1 (likes). We assume binarised ratings of 0 for films which users have not previously rated.

The probabilistic graphical model of the full generative distribution is given in Fig 6. We use $n$ to index films, and $m$ to index users. Each film has a known feature vector, $\mathbf{x}_n$, while each user has a latent weight-vector, $\mathbf{z}_m$, of the same length, describing whether or not they like any given feature. There are 18 features, indicating which genre tags the film has (Action, Adventure, Animation, Childrens,...). Each film may have more than one tag. The probability of the user liking a film is given by taking the dot-product of the film's feature vector with the latent weight-vector, and applying a sigmoid ($\sigma(\cdot)$) (final line)

$$
\begin{aligned}
\mathrm{P}\left(\boldsymbol{\mu}\right) &= \mathcal{N}(\boldsymbol{\mu}; \mathbf{0}_{18}, \mathbf{I}), \\
\mathrm{P}\left(\boldsymbol{\psi}\right) &= \mathcal{N}(\boldsymbol{\psi}; \mathbf{0}_{18}, \mathbf{I}), \\
\mathrm{P}\left(\mathbf{z}_m | \boldsymbol{\mu}, \boldsymbol{\psi}\right) &= \mathcal{N}(\mathbf{z}_m; \boldsymbol{\mu}, \exp(\boldsymbol{\psi})\mathbf{I}), \ m = 1, \ldots, M \\
\mathrm{P}\left(\mathrm{Rating}_{mn} | \mathbf{z}_m, \mathbf{x}_n\right) &= \mathrm{Bernoulli}(\mathrm{Rating}_{mn}; \sigma(\mathbf{z}_m^{\mathsf{T}} \mathbf{x}_n)), \ n = 1, \ldots, N
\end{aligned}
\tag{70}
$$

Additionally, we have latent vectors for the global mean, $\boldsymbol{\mu}$, and variance, $\boldsymbol{\psi}$, of the weight vectors.

The factorised proposal distribution $Q$ (also used as the initial approximate posterior for experiments using iterative methods) is given by

$$
\begin{aligned}
\mathrm{Q}\left(\boldsymbol{\mu}\right) &= \mathcal{N}(\boldsymbol{\mu}; \mathbf{0}_{18}, \mathbf{I}), \\
\mathrm{Q}\left(\boldsymbol{\psi}\right) &= \mathcal{N}(\boldsymbol{\psi}; \mathbf{0}_{18}, \mathbf{I}), \\
\mathrm{Q}\left(\mathbf{z}_m\right) &= \mathcal{N}(\mathbf{z}_m; \mathbf{0}_{18}, \mathbf{I}), \ m = 1, \ldots, M
\end{aligned}
\tag{71}
$$

We use a random subset of $N = 20$ films and $M = 450$ users for our experiment to ensure high levels of uncertainty. We use equally sized but disjoint subset of users held aside for calculation of the predictive log-likelihood.

---

[3]Dataset + License: `files.grouplens.org/datasets/movielens/ml-latest-small-README.html`

## C.4 OCCUPANCY DATASET

Occupancy models aim to infer the the true presence of a bird at a given observation site from repeated samples. The nature of collecting bird occupancy data for detection-nondetection datasets means that false detection or nondetection must be accounted for [Doser et al., 2022]. We fit a modified multi-species occupancy model to the north American bird breeding survey data[4]. This dataset records over 700 species of bird, along thousands of randomly selected road-side routes with readings taken at half mile intervals along the 24.5 mile routes for a total of 50 readings. The dataset covers the contiguous United States and the 13 provinces and territories of Canada.

The readings are taken during peak breeding season, usually in June, once per year. The presently available dataset covers the years 1966-2021, except for the year 2020. For our purposes we take the unit step function of the sum of every 10 readings to give R = 5 repeated samples from each route. Each route in each year has associated weather and quality covariates which are used in our model. We take a subset of $J = 12$ bird species, $M = 6$ years and $I = 200$ routes to form a training set on which to run our experiments, with a distinct test set sampling the same values of $J$ and $M$, but a different set of $I = 100$ routes. The $\text{Weather}_{jmi}$ covariate is the temperature at site $i$ on year $m$ replicated for each bird species. The $\text{Quality}_{jmi}$ covariate is indicates whether a particular series of readings at site $j$ on year $m$ followed all the recommended guidelines for recording birds.

We model the recording of a bird along a particular route in a particular repeated measurement as arising from a Bernoulli distribution, with logits given by the product of the weighted quality covariate and an inferred latent variable $z_{jmi}$ that indicates the true presence of a particular bird species $j$ along route $i$ in year $m$, as well as including some probability of a false-positive bird sighting. We model this latent variable $z_{mji}$ as also arising from a Bernoulli distribution, with logits given by weighted Weather covariates multiplied by a variable $\text{BirdYearMean}_{jm}$ representing the mean frequency of a specific species in a given year. The prior for the Weather and Quality weights, $\text{WeatherWeight}_{jmi}$ and $\text{QualityWeight}_{jmi}$ respectively, are normal with mean and log-variance sampled from standard normal priors. The variable $\text{BirdYearMean}_{jm}$ also has an hierarchical prior: first we sample a $\text{BirdMean}_{j}$ whose mean and log-variance are sampled from standard normal priors, then for each year we sample a $\text{BirdYearMean}_{jm}$ for each year and bird species from a normal distribution with mean $\text{BirdMean}_{j}$ and unit variance. The full model may be written as

$$
\begin{aligned}
\text{P}\left(\mu_{\text{BirdMean}}\right) &= \mathcal{N}(\mu_{\text{BirdMean}}; 0, 1), \\
\text{P}\left(\sigma_{\text{BirdMean}}\right) &= \mathcal{N}(\sigma_{\text{BirdMean}}; 0, 1), \\
\text{P}\left(\mu_{\text{QualityWeight}}\right) &= \mathcal{N}(\mu_{\text{QualityWeight}}; 0, 1), \\
\text{P}\left(\sigma_{\text{QualityWeight}}\right) &= \mathcal{N}(\sigma_{\text{QualityWeight}}; 0, 1), \\
\text{P}\left(\mu_{\text{WeatherWeight}}\right) &= \mathcal{N}(\mu_{\text{WeatherWeight}}; 0, 1), \\
\text{P}\left(\sigma_{\text{WeatherWeight}}\right) &= \mathcal{N}(\sigma_{\text{WeatherWeight}}; 0, 1), \\
\text{P}\left(\text{QualityWeight}_{j} \middle| \mu_{\text{QualityWeight}}, \sigma_{\text{QualityWeight}}\right) &= \mathcal{N}(\text{QualityWeight}_{j}; \mu_{\text{QualityWeight}}, \exp(\sigma_{\text{QualityWeight}})), \\
& \quad j \in \{1, ..., J\} \\
\text{P}\left(\text{WeatherWeight}_{j} \middle| \mu_{\text{WeatherWeight}}, \sigma_{\text{WeatherWeight}}\right) &= \mathcal{N}(\text{WeatherWeight}_{j}; \mu_{\text{WeatherWeight}}, \exp(\sigma_{\text{WeatherWeight}})), \\
& \quad j \in \{1, ..., J\} \\
\text{P}\left(\text{BirdMean}_{j} \middle| \mu_{\text{BirdMean}}, \sigma_{\text{BirdMean}}\right) &= \mathcal{N}(\text{BirdMean}_{j}; \mu_{\text{BirdMean}}, \exp(\sigma_{\text{BirdMean}})), \; j \in \{1, ..., J\} \\
\text{P}\left(\text{BirdYearMean}_{jm} \middle| \text{BirdMean}_{jm}\right) &= \mathcal{N}(\text{BirdYearMean}_{jm}; \text{BirdMean}_{jm}, 1), \; m \in \{1, ..., M\} \\
\text{logits}_{jmi}^{z} &= \text{BirdYearMean}_{jm} * \text{WeatherWeight}_{j} * \text{Weather}_{jmi}, \\
& \quad i \in \{1, ..., I\} \\
\text{P}\left(z_{jmi} \middle| \text{logits}_{jmi}^{z}\right) &= \text{Bernoulli}(z_{jmi}; \text{logits} = \text{logits}_{jmi}^{z}), \; i \in \{1, ..., I\} \\
\text{logits}_{jmir}^{y} &= z_{jmi} * \text{QualityWeight}_{j} * \text{Quality}_{jmir} + (1 - z_{jmi}) * (-10), \\
& \quad r \in \{1, ..., R\} \\
\text{P}\left(y_{jmir} \middle| \text{logits}_{jmir}^{y}\right) &= \text{Bernoulli}(y_{jmir}; \text{logits} = \text{logits}_{jmir}^{y}), r \in \{1, ..., R\}
\end{aligned}
$$

(72)

---

[4]Dataset: `https://www.sciencebase.gov/catalog/item/625f151ed34e85fa62b7f926`, licensing information is included with the dataset.

and the graphical model is presented in Fig. 7.

The factorised proposal distribution $Q$ is given by

$$
\begin{aligned}
\mathrm{Q}\left(\mu_{\mathrm{BirdMean}}\right) &= \mathcal{N}(\mu_{\mathrm{BirdMean}}; 0, 1), \\
\mathrm{Q}\left(\sigma_{\mathrm{BirdMean}}\right) &= \mathcal{N}(\sigma_{\mathrm{BirdMean}}; 0, 1), \\
\mathrm{Q}\left(\mu_{\mathrm{QualityWeight}}\right) &= \mathcal{N}(\mu_{\mathrm{QualityWeight}}; 0, 1), \\
\mathrm{Q}\left(\sigma_{\mathrm{QualityWeight}}\right) &= \mathcal{N}(\sigma_{\mathrm{QualityWeight}}; 0, 1), \\
\mathrm{Q}\left(\mu_{\mathrm{WeatherWeight}}\right) &= \mathcal{N}(\mu_{\mathrm{WeatherWeight}}; 0, 1), \\
\mathrm{Q}\left(\sigma_{\mathrm{WeatherWeight}}\right) &= \mathcal{N}(\sigma_{\mathrm{WeatherWeight}}; 0, 1), \\
\mathrm{Q}\left(\mathrm{QualityWeight}_j\right) &= \mathcal{N}(\mathrm{QualityWeight}_j; 0, 1), \ j \in \{1, ..., J\} \\
\mathrm{Q}\left(\mathrm{WeatherWeight}_j\right) &= \mathcal{N}(\mathrm{WeatherWeight}_j; 0, 1), \ j \in \{1, ..., J\} \\
\mathrm{Q}\left(\mathrm{BirdMean}_j\right) &= \mathcal{N}(\mathrm{BirdMean}_j; 0, 1), \ j \in \{1, ..., J\} \\
\mathrm{Q}\left(\mathrm{BirdYearMean}_{jm}\right) &= \mathcal{N}(\mathrm{BirdYearMean}_{jm}; 0, 1), \ m \in \{1, ..., M\} \\
\mathrm{logits}_{jmi}^z &= \mathrm{BirdYearMean}_{jm} * \mathrm{WeatherWeight}_j * \mathrm{Weather}_{jmi}, \ i \in \{1, ..., I\} \\
\mathrm{Q}\left(z_{jmi} \big| \mathrm{logits}_{jmi}^z\right) &= \mathrm{Bernoulli}(z_{jmi}; \mathrm{logits} = \mathrm{logits}_{jmi}^z), \ i \in \{1, ..., I\}
\end{aligned}
\tag{73}
$$

## C.5 EXPERIMENT DETAILS

In each experiment, we use a graphical model (specified above in Sections C.1-C.4) to define a prior/generative distribution and a proposal distribution over one of four datasets. In particular, the proposal distribution has the same structure as the prior, but with each latent variable parameterised independently of the others (cf. the dependencies shown between latents of the prior/generative distributions in Figs. 4-7).

In our first set of experiments (Figs. 1 and 2) we draw $K$ samples from the proposal and compute both the global and massively parallel estimators of the marginal likelihood (Eqs. 7 and 18). Taking the logarithm of these unbiased estimators of the marginal likelihood gives (via Jensen's inequality) a lower bound on the log-marginal likelihood, referred to as the ELBO.

With these $K$ samples ($K^N$ if we consider all possible combination of $n$-latent samples), we obtain 100 posterior samples of all latents via the importance sampling mechanism described in Section 3 (see Eq. 36). Then, we use these posterior latent samples to obtain predictive samples on an unseen test set of the data. We report the predictive log-likelihood of the test data given the predicted latent samples.

In our second set of experiments, we compare the 'one-shot' global and massively parallel importance sampling approaches with iterative methods—namely VI, IWAE and RWS. For each iteration of these methods, we obtain ELBOs and predictive log-likelihoods in the same manner as discussed above for global importance sampling, however, we also update the parameters of the proposal distribution at each iteration so as to maximise an objective function via the Adam optimiser. In the VI results, this objective function is the global ELBO, calculated with a single sample, i.e. $K = 1$. The IWAE results also use the global ELBO but with $K = 10$. The third method we consider is RWS in which we perform a maximum likelihood update of $Q$ using the posterior samples obtained by reweighting the proposal samples with the weights given in Eq. 6, also with $K = 10$.

We ran all experiments on an 80GB NVIDIA A100 GPU. For each experiment, we used global IS, MP IS, VI, IWAE, and RWS methods to obtain (for each value of $K$ in the IS methods, or each iteration of the iterative methods) 100 values of the ELBO, predictive log-likelihood (on a test set that was disjoint from the training set), which were averaged to produce the results presented in this work. For global IS and MP IS, we additionally ran a single warmup iteration before the final 100 to allow for memory management optimisation—the results of these warmup runs were then discarded. We present error bars in Figs. 1, and 2 representing the standard error over these 100 (post-warmup) runs.

We ran the globabl IS and MP IS experiments for $K \in \{3, 5, 10, 15, 20, 25, 30, 45, 60, 80, 10^2, 3 \times 10^2, 10^3, 3 \times 10^3, 10^4, 3 \times 10^4, 10^5, 3 \times 10^5, 10^6, 3 \times 10^6, 10^7\}$, however, only global IS on the chimpanzee dataset was able to run with each of these values (and no higher) — failures were mostly due to the large amount of memory required or due to numerical instability in the models.

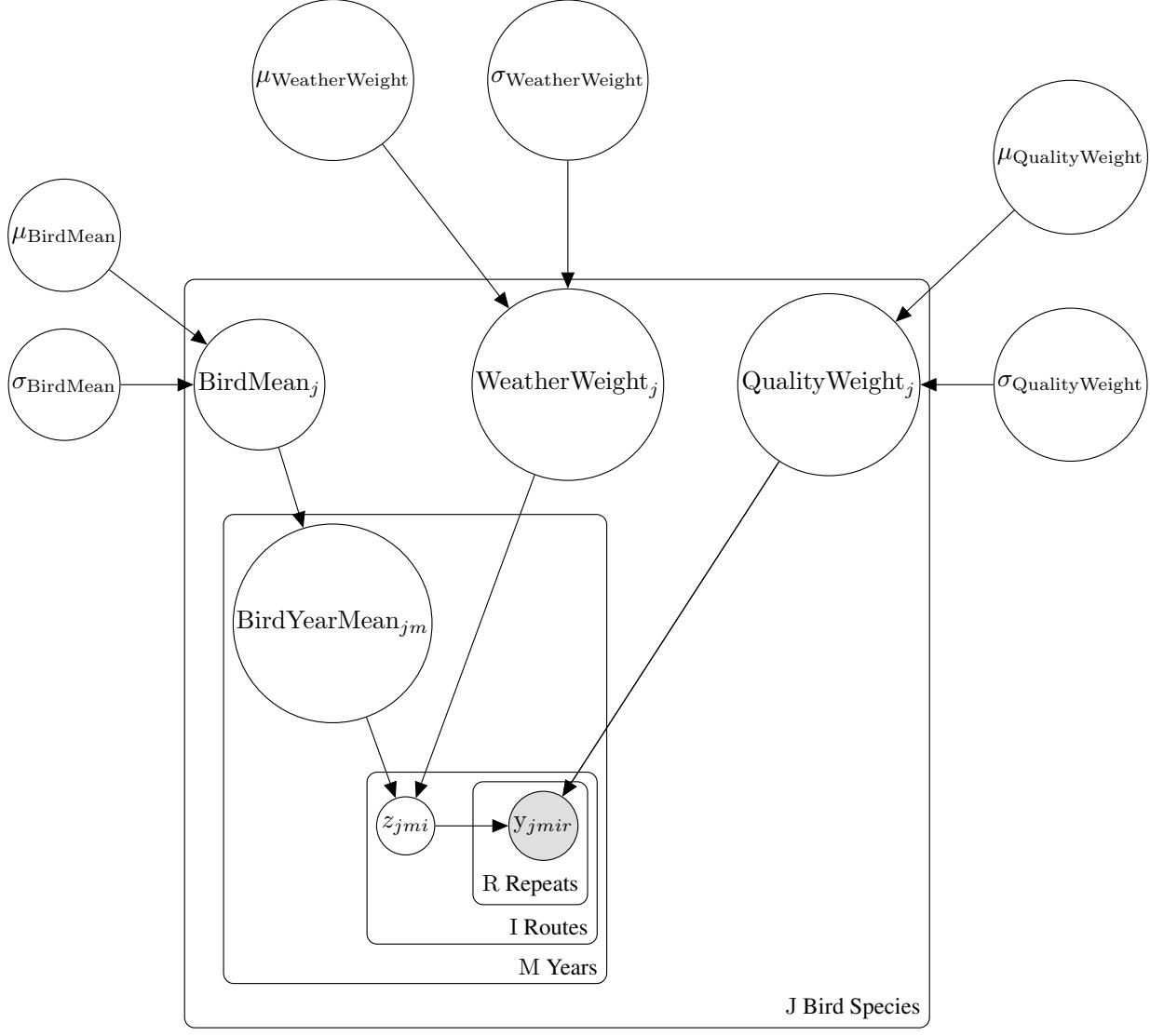

Figure 7: Graphical model for the Bird Occupancy dataset

For VI, IWAE and RWS, we optimized our approximate posterior using Adam with learning rates of $0.3, 0.1, 0.03, 0.01$ (we found that learning rates faster than $0.3$ were unstable on all models), and ran 100 optimization steps for both for each dataset/model. In Fig. 3 we plot only the best-performing learning rates for each method, where best-performing is decided as the a highest ELBO after a certain number of iterations, in particular we chose 75. In Appendix section D.1 we present plots of every learning rate for each method individually compared against MP IS and global IS on each model.

# D FURTHER RESULTS

## D.1 THE EFFECT OF LEARNING RATE ON ITERATIVE METHODS

Here we present plots of showing the results obtained on each model-method combination for every stable learning rate that we tried on the iterative models (excluding HMC). Note that no models were numerically stable with a learning rate higher than 0.3 for any of the iterative methods, and further that many of the following plots do not include learning rates of 0.3 or even 0.1 (for example MovieLens with RWS) as these were numerically unstable as well.

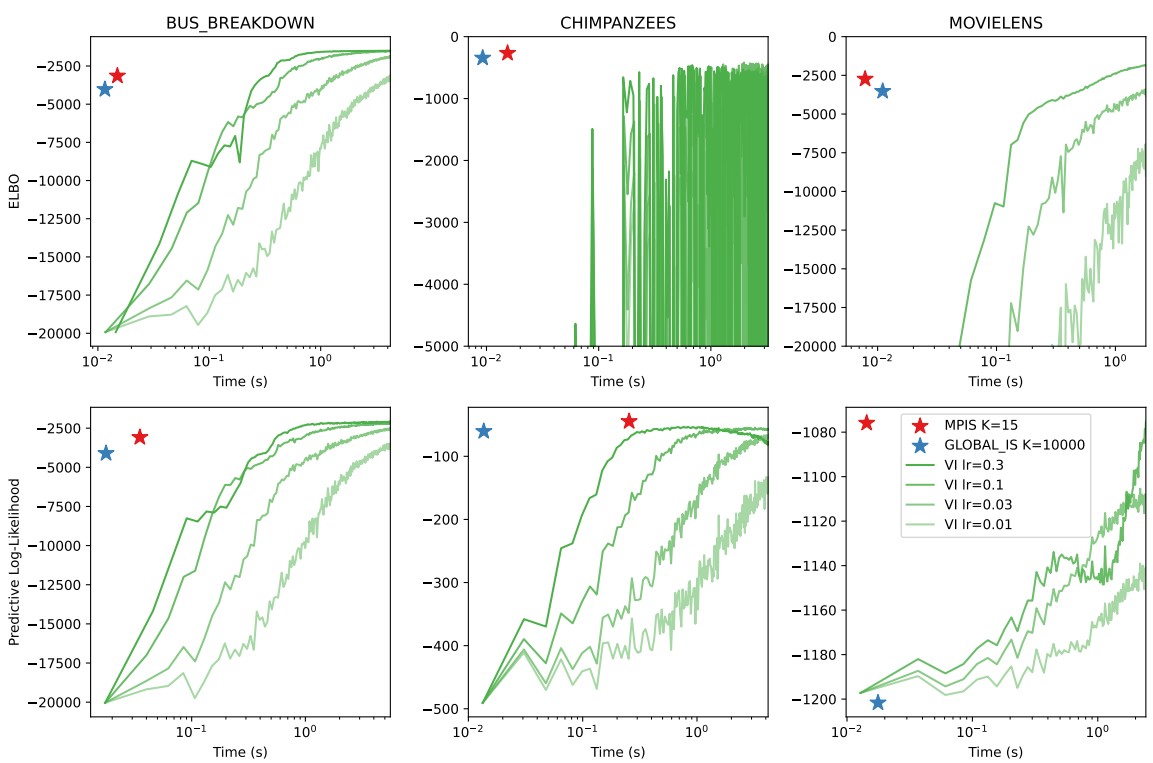

Figure 8: A comparison of results for VI on each dataset with varying learning rates.

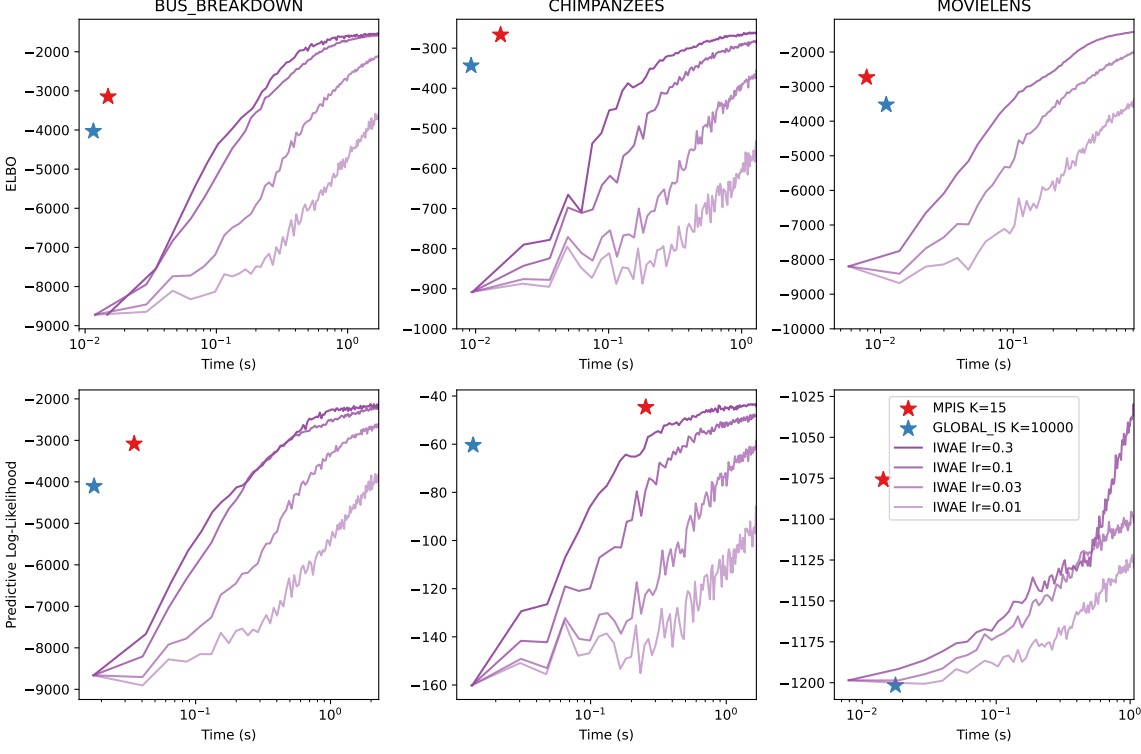

Figure 9: A comparison of results for IWAE on each dataset with varying learning rates.

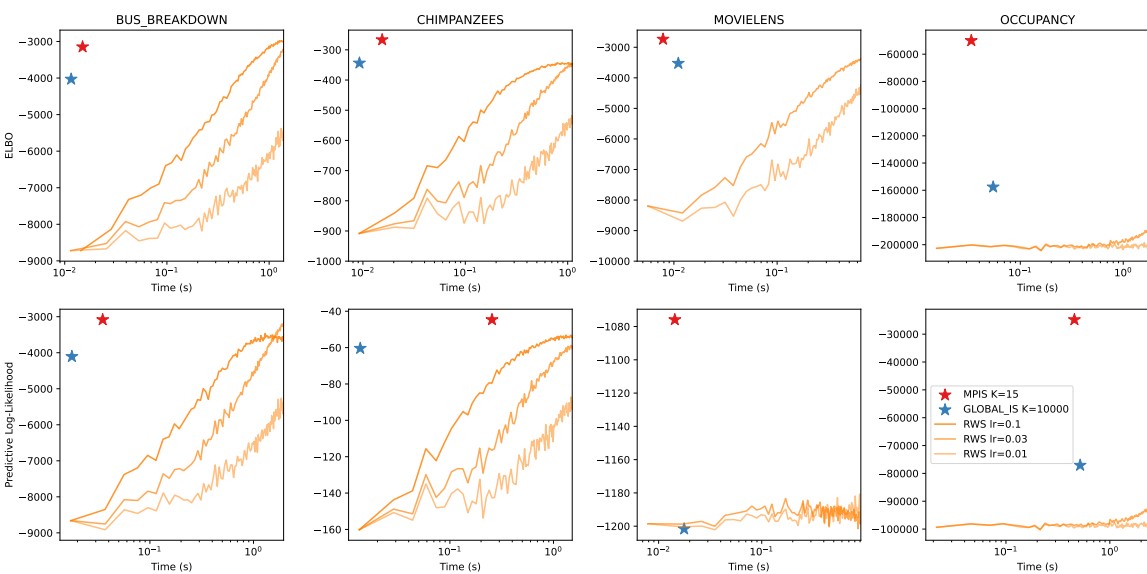

Figure 10: A comparison of results for RWS on each dataset with varying learning rates.