# OpenReview forum: "Using Autodiff to Estimate Posterior Moments, Marginals and Samples"
_auai.org/UAI/2024/Conference — UAI 2024 poster_

### Official Review · Reviewer_Ufcw · 2024-03-19

**Q2-1 Originality-Novelty:** 3
**Q2-2 Correctness-Technical Quality:** 3
**Q2-5 Clarity Of Writing:** 4

**Q1 Summary And Contributions:**

Proposes the 'source trick' as a method to estimate Bayesian marginal likelihoods and posterior moments using differentiation and facilitating massively parallel importance sampling

**Q2-3 Extent To Which Claims Are Supported By Evidence:**

3: Good: the main claims are supported by convincing evidence (in the form of adequate experimental evaluation, proofs, (pseudo-)code, references, assumptions).

**Q2-4 Reproducibility:**

2: Fair: key resources (e.g. proofs, code, data) are unavailable but key details (e.g. proof sketches, experimental setup) are sufficiently well-described for an expert to confidently reproduce the main results.

**Q3 Main Strengths:**

Very clear written providing an understandable introduction to some quite complex methodology.

Seemed to produce a compelling method to solve an interesting problem

**Q4 Main Weakness:**

I got lost in how the method is actually used. Section 4 appears to have developed a general method for posterior marginal likelihood and moment estimation and then Section 5 is talking about an ELBO for variational inference and a predictive log-likelihood and it is not easy to see how these two things are related, let alone if the experiments show that the method performs well. Part of the paper between Section 4 and 5 is missing

Estimators for the marginal likelihood and posterior moments were proposed but there was no mention of the variance of these estimators. This seems essential to these actually working in practice but there is no mention of this in the paper and no attempt to provide any theory for the variance.

**Q5 Detailed Comments To The Authors:**

Section 1

''... by exploiting conditional independence in the underlying graphical model'' could this be more specific. ''in the underlying graphical model if the latent variables $z$. Without this talk of a graphical mdoel comes from nowehere

Section 2

''Instead, this previous work simply takes the learned approximate posterior as an estimate of the true posterior, and does not attempt to correct for inevitable differences between approximate and true posterior''so your proposal is no longer unbiased? I found this confusing, VI and ELBO somewhat appears again in Section 5 and I got lost somehow. Could you introduce this part more clearly? (see main weakness)

Section 3

''Our goal is to compute the posterior distribution over latent variables'' Just to avoid any confusion, could you mention that these are sometimes referred to as parameters, then it is clear to everyone what you are doing.

''We often seek to obtain samples from the posterior or to compute posterior expectations,'' It would be more truthful to say that ''We often seek to compute posterior expectations... but these are usually intractable... and therefore we seek to sample...

''.. we draw a collection of $K$ samples from the full joint state space.'' this sounds confusing, ''the full joint state space of latent variables $z$''?

Below Equation (9) it might be worth pointing out that $m$ is the function whose expectation we wish to compute.

Figure 1 - ''The error bars represent the standard-error across 100 repeated experiments with different
random seeds'' is this 100 IS estimates for the same data. Or 100 resamples of the data. Ideally it would be good to assess the variance of your estimation procedure so consider repeated estimates for the same data

Section 5

Section 4 appeared to be presenting a general method for Importance Sampling but then Section 5 talks about an ELBO. It was not at all clear to me how these two were related. Could some more explanations be given here. It was also not clear to me how the ELBO and predictive log-likelihood values demonstrated that this importance sampling technique works somehow

Section 6

This is too short. What are the limitations of your approach? What future work could be done

**Q9 Complying With Reviewing Instructions:**

Yes

---

### Official Review · Reviewer_4kjz · 2024-03-20

**Q2-1 Originality-Novelty:** 4
**Q2-2 Correctness-Technical Quality:** 4
**Q2-5 Clarity Of Writing:** 3

**Q1 Summary And Contributions:**

The authors discuss the limitations of importance weighting in larger models and introduce a massively parallel importance sampling scheme that uses an exponential number of samples to compute posterior expectations, marginals, and samples. They highlight the challenges of exploiting conditional independencies due to complex algorithms required for backward traversals.

**Q2-3 Extent To Which Claims Are Supported By Evidence:**

3: Good: the main claims are supported by convincing evidence (in the form of adequate experimental evaluation, proofs, (pseudo-)code, references, assumptions).

**Q2-4 Reproducibility:**

2: Fair: key resources (e.g. proofs, code, data) are unavailable but key details (e.g. proof sketches, experimental setup) are sufficiently well-described for an expert to confidently reproduce the main results.

**Q3 Main Strengths:**

1.	The paper introduces a novel method for Bayesian inference that leverages massively parallel importance sampling and autodiff technology. This method addresses a significant challenge in scaling importance sampling to models with many latent variables.
2.	The authors highlight that their method avoids the need for complex backward traversals through the graphical model, which are typically required in traditional importance sampling schemes. This simplification can make the approach more accessible and easier to implement.
3.	By using massively parallel sampling, the proposed method can consider all possible combinations of latent variables efficiently, leading to potentially more accurate estimates of posterior moments, marginals, and samples.
4.	The method is not restricted to discrete graphical models but can operate with continuous latent variables and graphs with a mixture of continuous and discrete latent variables, broadening its applicability.

**Q4 Main Weakness:**

1.	As mentioned in the paper, the method can face memory constraints due to the large number of samples considered, which may limit its applicability in practice, especially on machines with limited memory. Can the authors give more information about this topic?
2.	While the paper compares the proposed method with existing iterative methods, it may not include a comprehensive comparison with the latest advancements in Bayesian inference techniques.
3.	The paper will be better to provide a detailed analysis of the method's scalability as the number of latent variables or the complexity of the model increases.

**Q5 Detailed Comments To The Authors:**

See q3 and q4

**Q9 Complying With Reviewing Instructions:**

Yes

---

> ### Author Rebuttal · Authors · 2024-04-05
>
> Thanks for your careful and positive review!
>
> >As mentioned in the paper, the method can face memory constraints due to the large number of samples considered, which may limit its applicability in practice, especially on machines with limited memory. Can the authors give more information about this topic?
>
> Yes, you can sometimes face memory constraints.
>
> In particular, the key issue is the tensors of log-probabilities.  If, for instance, you have $P(A)$ in the generative model, we will get a tensor with $K$ elements, for $P(B|A)$ we will get a tensor with $K^2$ elements, and for $P(C|A,B)$, we will get a tensor with $K^3$ elements.  There are two possible solutions to this problem:
> * First, while the tensors of log-probabilities have e.g.\ $K^2$ or $K^3$ elements, we only have $K$ samples for each latent variable.  We can therefore massively reduce memory consumption by lazily computing log-probability tensors only as they are necessary (matching modern approaches such as flash attention).
> * Second, grouping variables. If we grouped $A$ and $B$, that would mean we draw $K$ samples of $A,B$, and only consider K combinations (i.e.\ $A^{k=1},B^{k=1}$, $A^{k=2},B^{k=2}$, ..., $A^{k=K},B^{k=K}$ etc.), rather than the full $K^2$ combinations you would usually use.
> * Third, structuring your model. It turns out that in a hierarchical model, you can also reduce memory consumption dramatically by ensuring that variables in one plate depend only on other variables in the current plate, or on variables in the immediately preceding plate, and not on variables in higher-level plates.
>
> Given that these strategies are non-trivial in and of themselves (and therefore difficult to describe adequately in this response), we intend to include full details, along with an implementation as part of our ongoing work on integrating massively parallel inference with probabilistic programming.
>
>
>
> > The paper will be better to provide a detailed analysis of the method's scalability as the number of latent variables or the complexity of the model increases.
>
> In terms of dependency on the model structure, the situation is exactly analogous to the case of discrete graphical models, where the key quantity is the treewidth, which quantifies the degree to which conditional independencies in the model can be exploited to give efficient inference.
>
>
>
> > While the paper compares the proposed method with existing iterative methods, it may not include a comprehensive comparison with the latest advancements in Bayesian inference techniques.
>
> We have presented a comparison against modern inference methods including VI (Rezende et al. 2014; Kingma and Welling 2014), RWS (Bornschein et al. 2014) and IWAE (Burda et al. 2015). We'd be happy to run any additional baselines if you can let us know what methods you had in mind.

---

### Official Review · Reviewer_Kvga · 2024-03-23

**Q2-1 Originality-Novelty:** 2
**Q2-2 Correctness-Technical Quality:** 3
**Q2-5 Clarity Of Writing:** 1

**Q1 Summary And Contributions:**

Techniques are proposed for estimating marginal quantities associated with a directed graphical model, by approximating importance weighting over $K^n$ samples in only $O(Kn)$ computation. The main idea is to combine the source term trick (writing a marginalization over a posterior as the derivative of an expected exponential) with the massively parallel importance weighting proposed in [Heap et al., 2023]. Experiments show that these estimators are more efficient than basic importance sampling in time and number of samples needed to obtain similar estimates.

**Q2-3 Extent To Which Claims Are Supported By Evidence:**

2: Fair: the main claims are somewhat supported by evidence (but the experimental evaluation may be weak, or does not match entirely with the claims, important baselines may be missing, proofs contain important ideas but lack rigor, algorithmic details are only discussed superficially, references are imprecise, assumptions are not sufficiently motivated or explicated, etc.).

**Q2-4 Reproducibility:**

1: Poor: key details (e.g. proof sketches, experimental setup) are incomplete/unclear, or key resources (e.g. proofs, code, data) are unavailable.

**Q3 Main Strengths:**

- The proposed idea seems effective in giving importance-weighted estimates and samples in directed graphical models, addressing a major limitation in a clever way (building on the idea in [Heap et al.]).
- Evaluation on graphical models from diverse domains involving both continuous and discrete latent variables.

**Q4 Main Weakness:**

- The organization and clarity of the text can use substantial improvement:
  - The first two sections should be revised to more gently lead the reader into the setting and motivate the problem. It would have been very difficult to understand the first two paragraph had I not previously read [Heap et al., 2023].
    - For example, in the first paragraph, an expression involving $P$ and $Q$ is used before $P$ and $Q$ are defined, and $P(z|x)$ is in turn written before $z$ and $x$ are defined. This is hard on the reader.
    - Similarly, the second paragraph talked about an "underlying graphical model" (also not introduced).
  - The related work does not make sense without reading the rest of the paper. Each paragraph gives a long list of work without much commentary, then gives a somewhat rambling description of the benefits of the proposed approach, using details and language that were not introduced. I think this defeats the purpose of a related work section, which should summarize related work and show how it motivates the current one.
    - On this note, the interruptions of the flow of the rest of the paper with discussions of "what's different is..."  (after (39)) and "this is very different..." (after (27)) would better be made more concise and placed in a RW section placed nearer to the end of the paper.
  - I find the narrative style in the methods section to be distracting from the flow of the paper. It would have been clearer to state the d/dJ trick from section 2 without proof (it is not a difficult derivation and is fairly well known) and instead carefully state the assumptions (we assume there is a graphical model...) and the problem to be solved. Define relevant notation -- introducing things in a logical order and not ad hoc saves having to always write things like "remember ${\rm pa}(i)$ is" after (38) and again after (41).
    - The pseudocode in Appendix B helps, but does not make clear what each algorithm is actually meant to output and estimate.
- The experiments do not explain basic details of what was done, making it difficult to evaluate their results.
  - I presume that IW samples obtained using the proposed algorithm were used to train the graphical model's conditional distributions? This is not explained.
  - It is also not explained what exactly the quantities whose estimates are compared in the figures are, nor how the other methods (IWAE, RWS, VI -- by the way, what is "VI"? I guess it is variational inference, but there are many kinds of variational posteriors and ways of learning them...) can be used to estimate those quantities.
- There is no theoretical analysis of the algorithm, e.g., bounds on ESS. (Such bounds for the error of importance-weighted estimates exist in the case of all variables being independent, so we would hope for generalizations for nontrivial graphical models.)

**Q5 Detailed Comments To The Authors:**

See above.

**Q9 Complying With Reviewing Instructions:**

Yes

---

### Official Review · Reviewer_HkDR · 2024-03-25

**Q2-1 Originality-Novelty:** 4
**Q2-2 Correctness-Technical Quality:** 2
**Q2-5 Clarity Of Writing:** 3

**Q10 Ethical Concerns:**

No etical considerations are expected.

**Q1 Summary And Contributions:**

The paper proposes to use the "source term trick" to compute derived statistics, like expectations, marginal and conditional distribution when using importance weighting and in particular massive parallel method.

The paper provoides derivation of the three statistics and experiment for the complexity and accuracy evaluation of the approach.

**Q2-3 Extent To Which Claims Are Supported By Evidence:**

2: Fair: the main claims are somewhat supported by evidence (but the experimental evaluation may be weak, or does not match entirely with the claims, important baselines may be missing, proofs contain important ideas but lack rigor, algorithmic details are only discussed superficially, references are imprecise, assumptions are not sufficiently motivated or explicated, etc.).

**Q2-4 Reproducibility:**

4: Excellent: key resources (e.g. proofs, code, data) are available and key details (e.g. proof sketches, experimental setup) are comprehensively described for competent researchers to confidently and easily reproduce the main results.

**Q3 Main Strengths:**

1. A novel approach to compute expensive quantities from generative models.

2. Detail dertivation and justifications.

3. Multiple experiments and comparisons (Importance sampling, VI, IWAE, RWS).

**Q4 Main Weakness:**

1. disconnection between derivations and experiments
2. not clear description of which class of problem can benefit from this approach

**Q5 Detailed Comments To The Authors:**

1. the abstract is quite unclear, can be understood only after reading the paper, it would be nice to improve: "In principle, we can reason efficiently over Kn combinations of samples by exploiting conditional independencies in the generative model. However, in practice this requires complex algorithms that traverse backwards through the graphical model," not clear at all, even after reading the paper, why the backward direction is more difficult?
2. In the first paragraph, why the complexity is now exponential? Why the KL divergence is "proportional" to n? numerically? the KL measure the number of bit of information of the difference, so yes, it is proportional to the bit which possibly is the number of dimension. Tha paragraph is (interesting but) not clear.
3. also the third paragraph is not clear: "however...". “Worse, different traversals are needed for computing posterior expectations, marginals and samples, making a general implementation challenging” : why?
4. An important point, in the related work the authors mentioned that other approach are only limited to "discrete graphical model" which I interpreat as discrete variable graphical model. Why the approach of the author is more general?
5. in general would be nice to understand what type of problem can benefit from the proposed approach. When the authors describe the parent/graphical model are the authors considering the computational graph or the dependence among variables, are those dependence fix?
6. in equation (24), p_k does not depend on z. what is the reason?
7. From eq.(27) the authors describe the three class of statistics that can benefit from the "source trick" approach. Could you give example of statistics in relation to the introduction / background (e.g eq.1 and eq.2) or give some examples?
8. In the experiments the computed quantities are ELBO and "predictive" log-likelihood. Could you make more explicit what you are computing and how is related to the section 4. Otherwise the experiments are disconnected from the approach sections.
9. In figure 3, could you comment on the oscillation of the VI?
10. also for the other methods (GIS, VI, IWAE, RWS) would be nice to understand the setup of the experiments.
11. in general would be nice to understand the potential application and how to use the proposed approach in some concrete scenarios

**Q9 Complying With Reviewing Instructions:**

Yes

---

### Meta-Review · Area_Chair_dMSc · 2024-04-17

The paper introduces and interesting approach for using autodiff to estimate expectations in a massively parallell importance sampling context. While there wasn't a clear consensus, most reviewers favoured accept and the reviewer leaning reject had as main concerns writing. Reading the rebuttal and the review, I think most concerns have been resolved and can be implemented in a simple revision with requested changes and suggested improvements.